# Learning Topological Representations with Bidirectional Graph Attention Network for Solving Job Shop Scheduling Problem

**Cong Zhang**[1]     **Zhiguang Cao**[2]     **Yaoxin Wu**[3]     **Wen Song**[†,4]     **Jing Sun**[1]

[1]Nanyang Technological University, Singapore
[2]Singapore Management University, Singapore
[3]Department of Industrial Engineering & Innovation Sciences, Eindhoven University of Technology
[4]Institute of Marine Science and Technology, Shandong University, China
[1,4]{cong.zhang92@gmail.com, wensong@email.sdu.edu.cn}

## Abstract

Existing learning-based methods for solving job shop scheduling problems (JSSP) usually use off-the-shelf GNN models tailored to undirected graphs and neglect the rich and meaningful topological structures of disjunctive graphs (DGs). This paper proposes the topology-aware bidirectional graph attention network (TBGAT), a novel GNN architecture based on the attention mechanism, to embed the DG for solving JSSP in a local search framework. Specifically, TBGAT embeds the DG from a forward and a backward view, respectively, where the messages are propagated by following the different topologies of the views and aggregated via graph attention. Then, we propose a novel operator based on the message-passing mechanism to calculate the forward and backward topological sorts of the DG, which are the features for characterizing the topological structures and exploited by our model. In addition, we theoretically and experimentally show that TBGAT has linear computational complexity to the number of jobs and machines, respectively, strengthening our method's practical value. Besides, extensive experiments on five synthetic datasets and seven classic benchmarks show that TBGAT achieves new SOTA results by outperforming a wide range of neural methods by a large margin. All the code and data are publicly available online at https://github.com/zcaicaros/TBGAT.

## 1 INTRODUCTION

Production and logistics are integral components of contemporary manufacturing systems. Developing intelligent solutions for complex challenges in these domains through the application of machine learning (especially deep learning) techniques holds significant potential to advance the manufacturing industry and has become a topic of growing interest. Vehicle routing problems (VRP) Golden et al. [2008] in logistics have become increasingly popular. In contrast, the job shop scheduling problem (JSSP) Garey et al. [1976], which is more complex but has substantial applications in modern production systems, receives relatively less attention.

In most existing research on VRP, fully connected undirected graphs are commonly employed to model the interrelationships between nodes (customers and the depot). This graphical representation enables a range of algorithms to leverage graph neural networks (GNN) for learning problem representations, subsequently facilitating the resolution of VRP Kool et al. [2018], Xin et al. [2021], Joshi et al. [2022]. However, this densely connected topological structure is not applicable to JSSP, as it cannot represent the widespread precedence constraints among operations in a job. As a result, the mainstream research on JSSP utilizes disjunctive graphs (DG) Błażewicz et al. [2000], a sparse and directed graphical model, to depict instances and (partial) solutions. Presently, there are two emerging neural approaches for learning to solve JSSP based on disjunctive graphs. The first neural approach, predominantly featured in existing literature, focuses on learning construction heuristics, which adhere to a dispatching procedure that incrementally develops schedules from partial ones Zhang et al. [2020], Park et al. [2021b,a]. Nonetheless, this method is ill-suited for incorporating diverse work-in-progress (WIP) information (e.g., current machine load and job status) into the disjunctive graph Zhang et al. [2024]. The omission of such crucial information negatively impacts the performance of these construction heuristics. The second neural approach involves learning improvement heuristics for JSSP Zhang et al. [2024], wherein disjunctive graphs represent complete solutions to be refined, effectively transforming the scheduling problem into a graph optimization problem so as to bypass the issues faced by partial schedule representation.

---

[†]corresponding author.

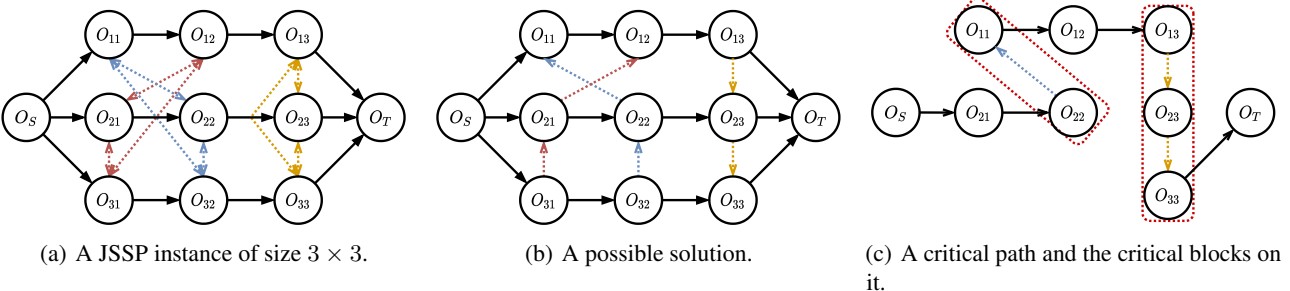

(a) A JSSP instance of size $3 \times 3$.

(b) A possible solution.

(c) A critical path and the critical blocks on it.

Figure 1: Disjunctive graph representations for JSSP instance and solution.

A prevalent approach in the aforementioned research involves utilizing canonical graph neural network (GNN) models, originally designed for undirected graphs, as the foundation for learning disjunctive graph embeddings. We contend that this approach may be inadvisable. Specifically, disjunctive graphs were initially introduced as a class of directed graphs, wherein the arc directions represent the processing order between operations. Notably, when modelling complete solutions, disjunctive graphs transform into directed acyclic graphs (DAGs), with their topological structures (node connectivity) exhibiting a bijective correspondence with the solution (schedule) space Błażewicz et al. [2000]. Learning these structures can substantially assist GNNs in acquiring in-depth knowledge of disjunctive graphs, enabling the differentiation between high-quality and inferior solutions. Nevertheless, conventional GNN models for undirected graphs lack the components necessary to accommodate these topological peculiarities during the learning of representations.

This paper introduces an end-to-end neural local search algorithm for solving JSSP using disjunctive graphs to represent complete solutions. We propose a novel bidirectional graph attention network (TBGAT) tailored for disjunctive graphs, effectively capturing their unique topological features. TB-GAT utilizes two independent graph attention modules to learn forward and backward views, incorporating forward and backward topological sorts. The forward view propagates messages from the root to the leaf, considering historical context, while the backward view propagates messages in the opposite direction, incorporating future schedule information. We show that forward topological order in disjunctive graphs corresponds to global processing orderings and present an algorithm for efficient GPU computation. For training, we design a deep reinforcement learning-based algorithm (DRL), particularly the REINFORCE algorithm with entropy regularization to train the TBGAT network. Theoretical analysis and experiments demonstrate TBGAT's linear computational complexity concerning the number of jobs and machines, a key attribute for practical JSSP solving.

We evaluate our proposed method against a range of neural approaches for JSSP, utilizing five synthetic datasets and seven classic benchmarks. Comprehensive experimental results demonstrate that our TBGAT model attains new state-of-the-art (SOTA) performance across all datasets, significantly surpassing all neural baselines.

## 2 RELATED LITERATURE

The rapid advancement of artificial intelligence has spurred a growing interest in addressing scheduling-related problems from a machine learning (especially the deep learning) perspective Dogan and Birant [2021]. For JSSP, the neural methods based on deep reinforcement learning (DRL) has emerged as the predominant machine learning paradigm. The majority of existing neural methods for JSSP are construction heuristics that sequentially construct solutions through a decision-making process. L2D Zhang et al. [2020] represents a seminal and exemplary study in this area, wherein a GIN-based Xu et al. [2019] policy learns latent embeddings of partial solutions, represented by disjunctive graphs, and selects operations for dispatch to corresponding machines at each construction step. A similar dispatching procedure is observed in RL-GNN Park et al. [2021b] and ScheduleNet Park et al. [2021a]. To incorporate machine status in decision-making, RL-GNN and ScheduleNet introduce artificial machine nodes with machine-progress information into the disjunctive graph. These augmented disjunctive graphs are treated as undirected, and a type-aware GNN model with two independent modules is proposed for extracting machine and task node embeddings. Despite considerable improvements over L2D, the performance remains suboptimal. DGERD Chen et al. [2022] follows a procedure similar to L2D but with a Transformer-based embedding networkVaswani et al. [2017]. A recent work, MatNet Kwon et al. [2021], employs an encoding-decoding framework for learning construction heuristics for flexible flow shop problems; however, its assumption of independent machine groups for operations at each stage is overly restrictive for JSSP. JSSenv Tassel et al. [2021] presents a carefully designed and well-optimized simulator for JSSP, extending from the OpenAI gym environment suite Brockman et al. [2016]. Rather than utilizing disjunctive graphs,

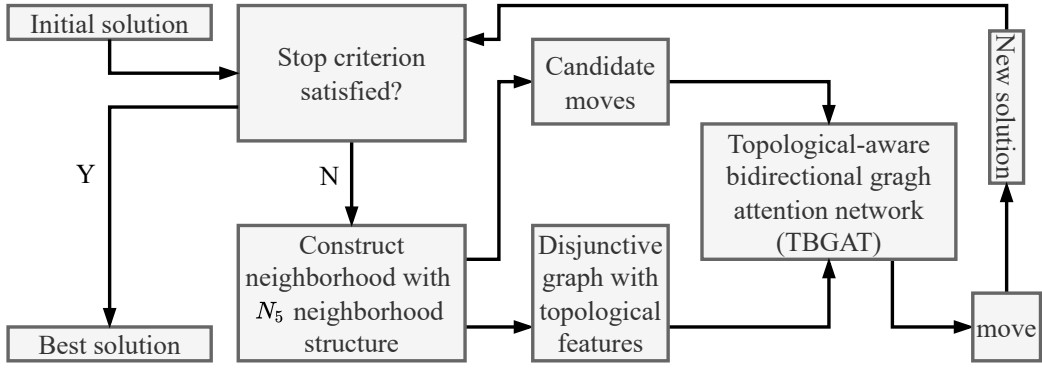

Figure 2: The local search procedure with TBGAT network.

JSSenv models and represents partial schedule states using Gantt charts Jain and Meeran [1999], and also proposes a DRL agent to solve JSSP instances individually in an online fashion. However, its online nature, requiring training for each instance, results in slower computation than offline-trained methods.

L2S Zhang et al. [2024] significantly narrows the optimality gaps by learning neural improvement heuristics for JSSP, thereby transforming the scheduling problem into a graph structure search problem. Specifically, L2S employs a straightforward local search framework in which a GNN-based agent learns to select pairs of operations for swapping, thus yielding new solutions. The GNN architecture comprises two modules based on GIN Xu et al. [2019] and GAT Veličković et al. [2018], focusing on the disjunctive graph and its subgraphs with different contexts separately. This design introduces two potential issues. Firstly, it is unclear whether GIN can maintain the same discriminative power for directed graphs as for undirected graphs. Secondly, the GAT network cannot allocate distinct attention scores to different neighbours during the representation learning since each node possesses only a single neighbour in either context subgraph, thus rendering the attention mechanism ineffective.

Existing attention-based GNN variants are readily available to embed DGs, e.g., Wang et al. [2019]. However, they either adopt a random-walk-based approach for aggregating the neighbourhood information Iyer et al. [2021], which neglects the precedent constraints and machine processing orders leading to inferior performance, or the computational complexity is not linear (w.r.t. the number of jobs and machines), making them less suitable for JSSP.

# 3 PREREQUISITE

## 3.1 THE JOB SHOP SCHEDULING PROBLEM.

A JSSP instance of size $|\mathcal{J}| \times |\mathcal{M}|$ comprises a set of jobs $\mathcal{J}$ and a set of machines $\mathcal{M}$. Each job $j \in \mathcal{J}$ must be pro-

cessed by each machine $m \in \mathcal{M}$ following a predefined order $O_{j1} \to \cdots \to O_{ji} \to \cdots \to O_{j|\mathcal{M}|}$, where $O_{ji} \in \mathcal{O}$ represents the $i$th operation of job $j$. Each operation $O_{ji}$ is allocated to a machine $m_{ji}$ with a processing time $p_{ji} \in \mathbb{N}$. Let $\mathcal{O}_j$ and $\mathcal{O}_m$ denote the collections of all operations for job $j$ and machine $m$, respectively. The operation $O_{ji}$ can be processed only when all its preceding operations in $\{O_{jk}|k < i\} \subset \mathcal{O}_j$ have been completed, which constitutes the precedent constraint. The objective is to identify a schedule $\eta : \mathcal{O} \to \mathbb{N}$, i.e., the starting time for each operation, that minimizes the makespan $C_{\max} = \max(\eta(O_{ij}) + p_{ij})$ without violating the precedent constraints.

## 3.2 THE DISJUNCTIVE GRAPH REPRESENTATION.

Disjunctive graphs Błażewicz et al. [2000] can comprehensively represent JSSP instances and solutions. As illustrated in Fig. 1(a), a $3 \times 3$ JSSP instance is represented by its corresponding disjunctive graph $G = \langle \mathcal{O}, \mathcal{C}, \mathcal{D} \rangle$. The artificial operations $O_S \in \mathcal{O}$ and $O_T \in \mathcal{O}$, which possess zero processing time, denote the start and end of the schedule, respectively. The solid arrows represent *conjunctions* ($\mathcal{C}$), which indicate precedent constraints for each job. The two-headed arrows signify *disjunctions* ($\mathcal{D}$) that mutually connect all operations belonging to the same machine, forming machine cliques with distinct colors. Discovering a solution is tantamount to assigning directions to disjunctive arcs such that the resulting graph is a directed acyclic graph (DAG) Balas [1969]. For instance, a solution to the JSSP instance in Fig. 1(a) is presented in Fig. 1(b), where the directions of all disjunctive arcs are determined, and the resulting graph is a DAG. Fig. 1(c) emphasizes a critical path of the solution in Fig. 1(b), i.e., the longest path from the source node $O_S$ to the sink node $O_T$, with critical blocks denoted by red frames (by longest we mean the total processing time of all operations along the path, except $O_S$ and $O_T$, is the largest among all paths from $O_S$ to $O_T$). The critical blocks are groups of operations belonging to the same machine on a critical path. The sum of the process-

ing times of operations along the critical path represents the makespan of the solution. Identifying a solution with a smaller makespan is equivalent to finding a disjunctive graph with a shorter critical path.

# 4 THE LOCAL SEARCH ALGORITHM WITH PROPOSED TBGAT NETWORK

We adopt a neural local search framework akin to L2S Zhang et al. [2024], as depicted in Fig. 2. The process commences with an initial solution produced by a construction heuristic (e.g., the dispatching rule), which is preserved as the current seed and the incumbent (the best-so-far solution). Subsequently, the disjunctive graph representation of the present seed is constructed, and the candidate moves (neighbours) are determined by employing the $N_5$ neighbourhood structure Nowicki and Smutnicki [1996]. This structure generates a candidate solution by exchanging the first or last pair of operations in a critical block along a critical path of the disjunctive graph, as demonstrated in Fig. 3. Following Nowicki and Smutnicki [1996], the first pair of operations in the initial critical block and the last pair of operations in the final critical block are excluded. Furthermore, in the presence of multiple critical paths, a random critical path is chosen Zhang et al. [2024], Nowicki and Smutnicki [1996]. The TBGAT network subsequently ingests the disjunctive graph of the current seed as input and produces one of the candidate moves. Ultimately, a new seed solution is acquired by swapping the operations in the disjunctive graph according to the selected operation pair, superseding the incumbent if superior. This search procedure persists until a stopping criterion is met, such as reaching a predetermined horizon, e.g., 5000 steps.

The aforementioned local search procedure can be recast in the framework of the Markov decision process (MDP). Specifically, the state, action, reward, and state transition are delineated as follows.

**State.** The state $s_t$ at time step $t$ represents the disjunctive graph representation of the seed solution at $t$. **Action.** The action set $A_t$ at $t$ comprises the candidate moves of $s_t$ calculated by applying the $N_5$ neighborhood structure. It is worth noting that $A_t$ may be dynamic and contingent on different seed solutions, where $A_t = \emptyset$ indicates that the current seed solution is the optimal one Nowicki and Smutnicki [1996]. **Reward.** The step-wise reward between any two consecutive states $s_t$ and $s_{t+1}$ is computed as $r(s_t, a_t) = \max(C_{\max}(s^*) - C_{\max}(s_{t+1}), 0)$, with $s^*$ denoting the incumbent. This is well-defined, as maximizing the cumulative reward is tantamount to maximizing the improvement to the initial solution, since $\sum_t^T r_t = C_{\max}(s_0) - C_{\max}(s_T)$. **State transition.** The state $s_t$ deterministically transits to the subsequent state $s_{t+1}$ by executing the selected action $a_t \in A_t$ at $s_t$, i.e., exchanging the operation pair in $s_t$. The episode terminates if $A_t = \emptyset$, beyond which the state

transition is ceased.

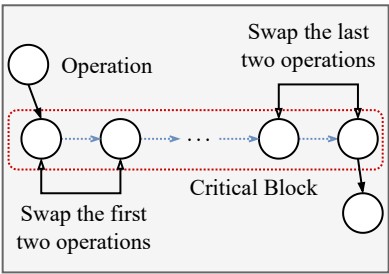

Figure 3: The $N_5$ neighborhood structure.

## 4.1 THE FORWARD AND BACKWARD VIEWS OF DGS

Disjunctive graphs (DGs) constitute a specific class of DAGs, exhibiting unique features. Firstly, by adhering to the direction of conjunctive and disjunctive arcs, each node $O_{ji}$ (barring $O_S$ and $O_T$) has neighbours from two directions. Namely, the predecessors pointing to $O_{ji}$ and successors pointing from $O_{ji}$. Nevertheless, existing GNN-based models either neglect the latter neighbors Zhang et al. [2020, 2024] or fail to differentiate the orientations of neighbors Park et al. [2021b]. In contrast, we recognize that neighbours from both directions are crucial and contain complementary information. Secondly, any $O_{ji}$ in a schedule cannot commence processing earlier than its predecessors, owing to the precedent constraints and the determined machine processing order. The earliest timestamp at which $O_{ji}$ may begin without violating these constraints is defined as the earliest starting time $EST_{ji}$. However, $O_{ji}$ is not mandated to start precisely at $EST_{ji}$ if not delaying the overall makespan. Instead, it has the latest starting time, denoted as $LST_{ji}$. The $EST_{ji}$ and $LST_{ji}$ collectively determine a schedule, which can be calculated recursively from a *forward* and *backward* perspective of the disjunctive graph Jungnickel and Jungnickel [2005], respectively, as follows,

$$EST_{ji} = \max_{O_{nl} \in \mathcal{P}_{O_{ji}}} (EST_{nl} + p_{nl}), \quad (1)$$

$$LST_{ji} = \min_{O_{nl} \in \mathcal{S}_{O_{ji}}} (LST_{nl} - p_{nl}), \quad (2)$$

where $\mathcal{P}_{O_{ji}}$ and $\mathcal{S}_{O_{ji}}$ represent the sets of predecessors and successors of $O_{ji}$, respectively. In the forward perspective, the computation traverses each $O_{ji}$ by adhering to the directions of conjunctive and disjunctive arcs, which are inverted in the backward perspective. An illustration of the forward and backward perspectives of message flow is demonstrated in Fig. 4.

The processing order of operations intrinsically determines the quality of a schedule, as it defines the message flows

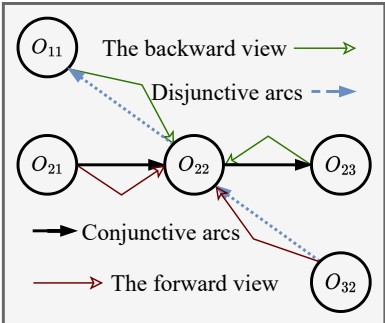

Figure 4: The forward and backward view of the DG.

of both forward and backward perspectives, represented by the connections of nodes and the orientations of arcs within the graph, i.e., the graph topology. Furthermore, a one-to-one correspondence exists between the space of disjunctive graph topologies and the space of feasible schedules. In other words, each pair of disjunctive graphs possessing distinct topologies corresponds to schedules of differing qualities. Hence, enabling the agent to leverage such topological features to learn discriminative embeddings for various schedules is highly advantageous, as it aids in distinguishing superior schedules from inferior ones. To achieve this, we employ the topological sort Wang et al. [2009], a partial order of nodes in a DAG that depicts their connectivity dependencies, as the topological features. The formal definition of the topological sort of nodes in the disjunctive graph is provided in Definition 1 below.

**Definition 1.** *(topological sort) Given any disjunctive graph $G = \langle \mathcal{O}, \mathcal{C}, \mathcal{D} \rangle$, there is a topological sort $\Phi : \mathcal{O} \to \mathbb{Z}$ such that for any pair of operations $O$ and $O'$ if there is an arc (disjunctive or conjunctive) connecting them as $O \to O'$, then $\Phi(O) < \Phi(O')$ must hold.*

Furthermore, within the context of JSSP, for any two operations $O, O' \in \mathcal{O}$, if $O$ is a prerequisite operation of $O'$, that is, $O$ must be processed prior to $O'$, then $O$ is required to have a higher ranking than $O'$ in the topological sort. This demonstrates that the topological sort serves as an alternative representation depicting the processing orders defined by the precedent constraints and the processing sequence of machines. Formally,

**Lemma 1.** *For any two operations $O_{ji}, O_{mk} \in \mathcal{O}$, if $O_{ji}$ is a prerequisite operation of $O_{mk}$, then $\overrightarrow{\Phi}(O_{ji}) < \overrightarrow{\Phi}(O_{mk})$ and $EST_{ji} < EST_{mk}$, where $\overrightarrow{\Phi} : \mathcal{O} \to \mathbb{Z}$ is the topological sort calculated from the **forward** view of the disjunctive graph.*

The proof is in Appendix A. A parallel conclusion can be derived for the backward view of the disjunctive graph, as presented below.

**Corollary 1.** *For any two operations $O_{ji}, O_{mk} \in \mathcal{O}$, if*

$O_{ji}$ *is a prerequisite operation of $O_{mk}$, then $\overleftarrow{\Phi}(O_{ji}) > \overleftarrow{\Phi}(O_{mk})$ and $LST_{ji} > LST_{mk}$, where $\overleftarrow{\Phi} : \mathcal{O} \to \mathbb{Z}$ is the topological sort calculated from the **backward** view of the disjunctive graph.*

The proof is in Appendix B. Our GNN model leverages forward and backward topological sorts to learn latent representations of disjunctive graphs. These sorts capture graph topology and global processing orders (Lemma 1 and Corollary 1). However, traditional algorithms for these sorts pose challenges for DRL agent training due to batch processing complexities, GPU-CPU communication overhead, and DRL data inefficiency. To overcome these issues, we propose MPTS, a novel algorithm based on a message-passing mechanism inspired by GNN computation. MPTS facilitates batch computation of forward and backward topological sorts, ensuring compatibility with GPU processing for more efficient training.

In fact, MPTS is universally applicable to any directed acyclic graph (DAG). Specifically, given a $DAG = \langle \mathcal{O}^*, \mathcal{E}^* \rangle$ with $\mathcal{O}^*$ and $\mathcal{E}^*$ denoting the sets of nodes and arcs, respectively. Let $\bar{\mathcal{O}} \subset \mathcal{O}^*$ be the set of nodes with zero in-degrees, and $\tilde{\mathcal{O}} = \mathcal{O}^* \setminus \bar{\mathcal{O}}$ be the set of the remaining nodes, where $\tilde{\mathcal{O}}' \subset \tilde{\mathcal{O}}$ is the set of nodes with zero out-degrees. We assign a message $m_x^t$ to each node $O_x \in \mathcal{O}^*$, initialized as $m_{\bar{x}}^0 = 1$ for nodes $O_{\bar{x}} \in \bar{\mathcal{O}}$ and $m_{\tilde{x}}^0 = 0$ for nodes $O_{\tilde{x}} \in \tilde{\mathcal{O}}$. Next, we define a message-passing operator $MPO : m_x^t \to m_x^{t+1}$, which calculates and updates the message for each node $O_x$ as $m_x^{t+1} = \max_{O_y \in \mathcal{N}_x}(m_y^t)$, where $\mathcal{N}_x$ is a neighborhood of $O_x$ containing all nodes $O_y$ pointing to $O_x$. Finally, let $L$ denote the length of the global longest path in $DAG$, and $L_x$ represent the length of any longest path from nodes in the set $O_{\bar{x}}$ to $O_x$, we can demonstrate the following.

**Theorem 1.** *After applying MPO for $L_x$ times, $m_x^{L_x} = 1$ for all nodes $O_x \in \mathcal{O}$. Moreover, for any pair of nodes $O_x$ and $O_z$ connected by a path, if $L_x < L_z \leq L$ then $\Phi(O_x) < \Phi(O_z)$.*

The proof is in Appendix C. Theorem 1 suggests an efficient method for computing the topological sort, in which we can iteratively apply the MPO on any $DAG$ and gather the nodes $O_x$ with $m_x^t = 1$ at each iteration $0 \leq t \leq |\mathcal{O}^*|$. Consequently, nodes collected in earlier iterations must hold higher ranks than those in later iterations within the topological sort. Since disjunctive graphs also belong to the class of DAGs, Theorem 1 can be directly applied to compute the forward topological sort $\overrightarrow{\Phi}$, as stated in Lemma 1, and the backward topological sort $\overleftarrow{\Phi}$, as indicated in Corollary 1, respectively. Given that the MPO operator can be readily implemented on a GPU to leverage its powerful parallel computation capabilities, MPO is anticipated to be more efficient than traditional algorithms when handling multiple disjunctive graphs concurrently. To substantiate

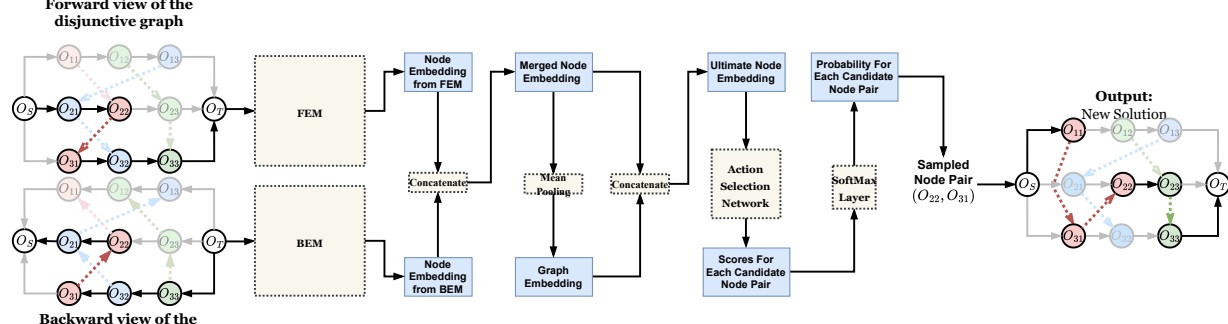

Figure 5: The architecture of the policy network.

this assertion, we provide an empirical comparison in the experimental section.

## 4.2 GRAPH EMBEDDING WITH TBGAT

In order to effectively learn graph embeddings by leveraging the topological features of disjunctive graphs, we introduce a novel bidirectional graph attention network designed to embed the forward and backward views of the disjunctive graph using two independent modules, respectively. For each view, message propagation adheres to the topology of respective view, and aggregation is accomplished through an attention mechanism, which distinguish our model from previous works. The overarching architecture of the proposed TBGAT is depicted in Fig. 5.

### 4.2.1 The forward embedding module

In the forward view of the disjunctive graph, each node $O_x \in \mathcal{O}$ is associated with a three-dimensional raw feature vector $\overrightarrow{\mathbf{h}}_x^0 = (p_x, est_x, \overrightarrow{\Phi}_x) \in \mathbb{R}^3$. Here, $p_x$ represents the processing time of node $O_x$, $est_x$ represents the earliest starting time, and $\overrightarrow{\Phi}_x$ represents the forward topological sort of node $O_x$. The forward embedding module (FEM) is a graph neural network that consists of $L$ layers. Each layer produces a new message for a node by aggregating the messages of this node and its neighbours from the previous layer. The aggregation operator used for updating the message of node $O_x$ is a weighted sum, where the weights are determined by attention scores that indicate the importance of each message. In other words, the aggregation operator for updating the message of node $O_x$ is expressed as follows,

$$\overrightarrow{\mathbf{h}}_x^{l+1} = \alpha_{x,x}^l \theta_{ag}^l \overrightarrow{\mathbf{h}}_x^l + \sum_{O_y \in \mathcal{N}(O_x)} \alpha_{x,y}^l \theta_{ag}^l \overrightarrow{\mathbf{h}}_y^l, 0 \leq l \leq L - 1.$$
(3)

The above attention scores $\alpha_{x,x}^l$ and $\alpha_{x,y}^l$ are used as weights in the aggregation operator for updating the message

of node $O_x$ in the forward embedding module (FEM). In specific, $\alpha_{x,x}^l$ and $\alpha_{x,y}^l$ are the attention scores for the messages of node $O_x$ itself and its neighbor $O_y$, respectively; $\theta_{ag}^l$ and $\mathcal{N}(x)$ denote the learnable parameters for the aggregation operator and the neighborhood of node $O_x$, respectively. Particularly, the two attention scores are computed through a widely-used practice in graph neural networks as follows,

$$\alpha_{x,*}^l = \frac{\exp\left(\text{LReLU}\left([\mathbf{a}^l]^\top [\theta_{at}^l \overrightarrow{\mathbf{h}}_x^l \| \theta_{at}^l \overrightarrow{\mathbf{h}}_*^l]\right)\right)}{\sum_{k \in \mathcal{N}'(O_x)} \exp\left(\text{LReLU}\left([\mathbf{a}^l]^\top [\theta_{at}^l \overrightarrow{\mathbf{h}}_x^l \| \theta_{at}^l \overrightarrow{\mathbf{h}}_k^l]\right)\right)},$$
(4)

where $\|$ denotes the concatenation operator; $*$ refers to $x$ or $y$; $\mathcal{N}'(O_x) = \mathcal{N}(O_x) \cup \{O_x\}$; LReLU is the LeakyRelU layer Radford et al. [2015]; and $\mathbf{a}^l$ and $\theta_{at}^l$ are the learnable parameters.

### 4.2.2 The backward embedding module

The backward embedding module (BEM) has an architecture that is similar to that of the forward embedding module. However, there is a key difference in the raw feature used for each node. Specifically, the raw feature for each node $O_x$ is substituted with the backward hidden state vector $\overleftarrow{\mathbf{h}}_x^0 = (p_x, lst_x, \overleftarrow{\Phi}_x) \in \mathbb{R}^3$, where $p_x$ represents the processing time of node $O_x$, $lst_x$ represents the latest starting time of node $O_x$, and $\overleftarrow{\Phi}_x$ represents the backward topological sort of node $O_x$. This allows the backward embedding module to encode the information about the temporal dependencies between nodes in the reverse order of the forward embedding module.

### 4.2.3 Merging the forward and backward embeddings

The merged embedding for a given node $O_x$ is derived by concatenating the output vector of the last layer of the forward embedding module (FEM) and the last layer of the backward embedding module (BEM), which results in a single vector that encodes both the forward and backward

Table 1: Performance on classic benchmarks. "Gap": the average gap to the best known solutions reported in the previous literature. "Time": the average time for solving a single instance in seconds ("s"), minutes ("m"), or hours ("h"). For the neural methods, the best results are marked in bold, where the global best results are colored blue.

| Method | Taillard | | | | | | | | ABZ | | FT | | |
|---|---|---|---|---|---|---|---|---|---|---|---|---|---|
| | 15×15 | 20×15 | 20×20 | 30×15 | 30×20 | 50×15 | 50×20 | 100×20 | 10×10 | 20×15 | 6×6 | 10×10 | 20×5 |
| | Gap Time | Gap Time | Gap Time | Gap Time | Gap Time | Gap Time | Gap Time | Gap Time | Gap Time | Gap Time | Gap Time | Gap Time | Gap Time |
| CP-SAT Perron and Furnon | 0.1% 7.7m | 0.2% 0.8h | 0.7% 1.0h | 2.1% 1.0h | 2.8% 1.0h | 0.0% 0.4h | 2.8% 0.9h | 3.9% 1.0h | 0.0% 0.8s | 1.0% 1.0h | 0.0% 0.1s | 0.0% 4.1s | 0.0% 4.8s |
| L2D Zhang et al. [2020] | 24.7% 0.4s | 30.0% 0.6s | 28.8% 1.1s | 30.5% 1.3s | 32.9% 1.5s | 20.0% 2.2s | 23.9% 3.6s | 12.9% 28.2s | 14.8% 0.1s | 24.9% 0.6s | 14.5% 0.1s | 21.0% 0.2s | 36.3% 0.2s |
| RL-GNN Park et al. [2021b] | 20.1% 3.0s | 24.9% 7.0s | 29.2% 12.0s | 24.7% 24.7s | 32.0% 38.0s | 15.9% 1.9m | 21.3% 3.2m | 9.2% 28.2m | 10.1% 0.5s | 29.0% 7.3s | 29.1% 0.1s | 22.8% 0.5s | 14.8% 1.3s |
| ScheduleNet Park et al. [2021a] | 15.3% 3.5s | 19.4% 6.6s | 17.2% 11.0s | 19.1% 17.1s | 23.7% 28.3s | 13.9% 52.5s | 13.5% 1.6m | 6.7% 7.4m | 6.1% 0.7s | 20.5% 6.6s | 7.3% 0.2s | 19.5% 0.8s | 28.6% 1.6s |
| L2S-500 Zhang et al. [2024] | 9.3% 9.3s | 11.6% 10.1s | 12.4% 10.9s | 14.7% 12.7s | 17.5% 14.0s | 11.0% 16.2s | 13.0% 22.8s | 7.9% 50.2s | 2.8% 7.4s | 11.9% 10.2s | 0.0% 6.8s | 9.9% 7.5s | 6.1% 7.4s |
| TBGAT-500 | 8.0% 12.6s | 9.9% 14.6s | 10.0% 17.5s | 13.3% 17.2s | 16.4% 19.3s | 9.6% 23.9s | 11.9% 24.4s | 6.4% 42.0s | 1.1% 9.2s | 11.8% 12.8s | 0.0% 7.4s | 5.2% 10.3s | 2.7% 11.7s |

| Method | LA | | | | | | | | SWV | | ORB | YN | |
|---|---|---|---|---|---|---|---|---|---|---|---|---|---|
| | 10×5 | 15×5 | 20×5 | 10×10 | 15×10 | 20×10 | 30×10 | 15×15 | 20×10 | 20×15 | 50×10 | 10×10 | 20×20 |
| | Gap Time | Gap Time | Gap Time | Gap Time | Gap Time | Gap Time | Gap Time | Gap Time | Gap Time | Gap Time | Gap Time | Gap Time | Gap Time |
| CP-SAT Perron and Furnon | 0.0% 0.1s | 0.0% 0.2s | 0.0% 0.5s | 0.0% 0.4s | 0.0% 21.0s | 0.0% 12.9m | 0.0% 13.7s | 0.0% 30.1s | 0.1% 0.8h | 2.5% 1.0h | 1.6% 0.5h | 0.0% 4.8s | 0.5% 1.0h |
| L2D Zhang et al. [2020] | 14.3% 0.1s | 5.5% 0.1s | 4.2% 0.2s | 21.9% 0.1s | 24.6% 0.2s | 24.7% 0.4s | 8.4% 0.7s | 27.1% 0.4s | 41.4% 0.3s | 40.6% 0.6s | 30.8% 1.2s | 31.8% 0.1s | 22.1% 0.9s |
| RL-GNN Park et al. [2021b] | 16.1% 0.2s | 1.1% 0.5s | 2.1% 1.2s | 17.1% 0.5s | 22.0% 1.5s | 27.3% 3.3s | 6.3% 11.3s | 21.4% 2.8s | 28.4% 3.4s | 29.4% 7.2s | 16.8% 51.5s | 21.8% 0.5s | 24.8% 11.0s |
| ScheduleNet Park et al. [2021a] | 12.1% 0.6s | 2.7% 1.2s | 3.6% 1.9s | 11.9% 0.8s | 14.6% 2.0s | 15.7% 4.1s | 3.1% 9.3s | 16.1% 3.5s | 34.4% 3.9s | 30.5% 6.7s | 25.3% 25.1s | 20.0% 0.8s | 18.4% 11.2s |
| L2S-500 Zhang et al. [2024] | 2.1% 6.9s | 0.0% 6.8s | 0.0% 7.1s | 4.4% 7.5s | 6.4% 8.0s | 7.0% 8.9s | 0.2% 10.2s | 7.3% 9.0s | 29.6% 8.8s | 25.5% 9.7s | 21.4% 12.5s | 8.2% 7.4s | 12.4% 11.7s |
| TBGAT-500 | 2.1% 2.3s | 0.0% 0.9s | 0.0% 1.7s | 1.8% 9.1s | 3.6% 10.8s | 5.0% 11.4s | 0.0% 4.9s | 5.5% 12.1s | 23.0% 15.6s | 23.7% 17.0s | 20.3% 29.8s | 7.0% 10.4s | 9.6% 14.3s |

Table 2: Generalization performance on classic benchmarks. "Gap": the average gap to the best known solutions reported in the previous literature. "Time": the average time for solving a single instance in seconds ("s"), minutes ("m"), or hours ("h"). For the neural methods, the best results are marked in bold, where the global best results are colored blue.

| Method | Taillard | | | | | | | | ABZ | | FT | | |
|---|---|---|---|---|---|---|---|---|---|---|---|---|---|
| | 15×15 | 20×15 | 20×20 | 30×15 | 30×20 | 50×15 | 50×20 | 100×20 | 10×10 | 20×15 | 6×6 | 10×10 | 20×5 |
| | Gap Time | Gap Time | Gap Time | Gap Time | Gap Time | Gap Time | Gap Time | Gap Time | Gap Time | Gap Time | Gap Time | Gap Time | Gap Time |
| CP-SAT Perron and Furnon | 0.1% 7.7m | 0.2% 0.8h | 0.7% 1.0h | 2.1% 1.0h | 2.8% 1.0h | 0.0% 0.4h | 2.8% 0.9h | 3.9% 1.0h | 0.0% 0.8s | 1.0% 1.0h | 0.0% 0.1s | 0.0% 4.1s | 0.0% 4.8s |
| L2S-1000 Zhang et al. [2024] | 8.6% 18.7s | 10.4% 20.3s | 11.4% 22.2s | 12.9% 24.7s | 15.7% 28.4s | 9.0% 32.9s | 11.4% 45.4s | 6.6% 1.7m | 2.8% 15.0s | 11.2% 19.9s | 0.0% 13.5s | 8.0% 15.1s | 3.9% 15.0s |
| L2S-2000 Zhang et al. [2024] | 7.1% 37.7s | 9.4% 41.5s | 10.2% 44.7s | 11.0% 49.1s | 14.0% 56.8s | 6.9% 65.7s | 9.3% 90.9s | 5.1% 3.4m | 2.8% 30.1s | 9.5% 39.3s | 0.0% 27.2s | 5.7% 30.0s | 1.5% 29.9s |
| L2S-5000 Zhang et al. [2024] | 6.2% 92.2s | 8.3% 1.7m | 9.0% 1.9m | 9.0% 2.0m | 12.6% 2.4m | 4.6% 2.8m | 6.5% 3.8m | 3.0% 8.4m | 1.4% 75.2s | 8.6% 99.6s | 0.0% 67.7s | 5.6% 74.8s | 1.1% 73.3s |
| TBGAT-1000 | 6.1% 24.9s | 8.7% 28.7s | 9.0% 34.1s | 10.9% 33.7s | 14.0% 37.3s | 7.5% 46.9s | 9.4% 47.5s | 4.9% 82.3s | 1.1% 17.9s | 10.1% 25.3s | 0.0% 14.2s | 4.8% 20.5s | 1.1% 23.2s |
| TBGAT-2000 | 5.1% 49.7s | 7.3% 56.5s | 7.9% 67.0s | 9.4% 65.9s | 12.1% 72.3s | 5.6% 92.9s | 7.5% 93.7s | 3.0% 2.7m | 1.1% 35.5s | 7.1% 50.0s | 0.0% 28.0s | 4.7% 45.8s | 0.8% 45.8s |
| TBGAT-5000 | 4.6% 2.1m | 6.3% 2.3m | 7.0% 2.7m | 6.8% 2.7m | 9.7% 2.9m | 2.6% 3.9m | 5.0% 3.9m | 1.2% 6.7m | 0.8% 88.2s | 6.6% 2.1m | 0.0% 69.4s | 2.9% 1.7m | 0.0% 1.9m |

| Method | LA | | | | | | | | SWV | | ORB | YN | |
|---|---|---|---|---|---|---|---|---|---|---|---|---|---|
| | 10×5 | 15×5 | 20×5 | 10×10 | 15×10 | 20×10 | 30×10 | 15×15 | 20×10 | 20×15 | 50×10 | 10×10 | 20×20 |
| | Gap Time | Gap Time | Gap Time | Gap Time | Gap Time | Gap Time | Gap Time | Gap Time | Gap Time | Gap Time | Gap Time | Gap Time | Gap Time |
| CP-SAT Perron and Furnon | 0.0% 0.1s | 0.0% 0.2s | 0.0% 0.5s | 0.0% 0.4s | 0.0% 21.0s | 0.0% 12.9m | 0.0% 13.7s | 0.0% 30.1s | 0.1% 0.8h | 2.5% 1.0h | 1.6% 0.5h | 0.0% 4.8s | 0.5% 1.0h |
| L2S-1000 Zhang et al. [2024] | 1.8% 14.0s | 0.0% 13.9s | 0.0% 14.5s | 2.3% 15.0s | 5.1% 16.0s | 5.7% 17.5s | 0.0% 20.4s | 6.6% 18.2s | 24.5% 17.6s | 23.5% 19.0s | 20.1% 25.4s | 6.6% 15.0s | 10.5% 23.4s |
| L2S-2000 Zhang et al. [2024] | 1.8% 27.9s | 0.0% 28.3s | 0.0% 28.7s | 1.8% 30.1s | 4.0% 32.2s | 3.4% 34.2s | 0.0% 40.4s | 6.3% 35.9s | 21.8% 34.7s | 21.7% 38.8s | 19.0% 49.5s | 5.7% 29.9s | 9.6% 47.0s |
| L2S-5000 Zhang et al. [2024] | 1.8% 70.0s | 0.0% 71.0s | 0.0% 73.7s | 0.9% 75.1s | 3.4% 80.9s | 2.6% 85.4s | 0.0% 99.3s | 5.9% 88.8s | 17.8% 86.9s | 17.0% 99.8s | 17.1% 2.1m | 3.8% 75.9s | 8.7% 1.9m |
| TBGAT-1000 | 1.6% 3.1s | 0.0% 1.9s | 0.0% 3.3s | 1.8% 18.2s | 3.5% 21.4s | 4.0% 22.8s | 0.0% 6.3s | 5.3% 24.5s | 19.2% 31.2s | 20.1% 34.3s | 18.5% 59.5s | 5.7% 20.8s | 8.0% 28.1s |
| TBGAT-2000 | 1.5% 4.8s | 0.0% 3.7s | 0.0% 6.5s | 1.6% 36.7s | 3.0% 42.4s | 2.9% 43.9s | 0.0% 8.9s | 5.1% 48.9s | 14.6% 61.7s | 18.1% 67.6s | 16.6% 2.0m | 5.1% 41.8s | 7.0% 55.8s |
| TBGAT-5000 | 1.5% 9.8s | 0.0% 9.2s | 0.0% 16.1s | 1.4% 93.1s | 2.4% 1.8m | 2.1% 52.0s | 0.0% 16.9s | 4.4% 2.0m | 11.1% 2.5m | 15.7% 2.7m | 14.5% 4.9m | 4.5% 1.7m | 5.7% 2.3m |

temporal dependencies of the node as

$$\mathbf{h}_x^L = \overrightarrow{\mathbf{h}}_x^L || \overleftarrow{\mathbf{h}}_x^L, \text{for all } O_x. \qquad (5)$$

Moreover, we concatenate $\mathbf{h}_x^L$ with the graph embedding to form the ultimate embedding for each node as

$$\mathbf{h}_x = \mathbf{h}_x^L || \mathbf{h}_G, \text{for all } O_x, \qquad (6)$$

where $\mathbf{h}_G$ is obtained with the mean pooling of the node embeddings as $\mathbf{h}_G = \frac{1}{|\mathcal{O}|} \sum_{O_x \in \mathcal{O}} \mathbf{h}_x$.

### 4.3 ACTION SELECTION

Given the node embeddings $\mathbf{h}_x$ and a graph embedding $\mathbf{h}_G$, we calculate a "score" for selecting each operation pair in the $N_5$ neighborhood structure as follows. For any pair of operations $(O_x, O_z)$ obtained from the $N_5$ neighborhood structure, we first concatenate the corresponding node embeddings $\mathbf{h}_x$ and $\mathbf{h}_z$ to obtain the joint representation $\mathbf{h}_{xz}$ for the action $(O_x, O_z)$. We then feed $\mathbf{h}_{xz}$ into the action selection network $Net_A$, i.e., a multi-layer perceptron (MLP) with $L_A$ hidden layers, to obtain a scalar score $sc_{xz}$ for the action $(O_x, O_z)$. The score $sc_{xz}$ is then normalized to obtain a probability $p_{xz}$, from which we could sample an action.

**Theorem 2.** *The TBGAT network has linear time complexity w.r.t both $|\mathcal{J}|$ and $|\mathcal{M}|$.*

By following the proof of Theorem 4.1 in paper Zhang et al. [2024], it is not difficult to see that the FEM module and the BEM modules pose linear computational complexity. Then, since the action selection network is an MLP that is, of course, linear to $|\mathcal{J}|$ and $|\mathcal{M}|$, TBGAT has linear computational complexity regarding the number of jobs $|\mathcal{J}|$ and the number of machines $|\mathcal{M}|$, respectively. $\square$

## 4.4 THE ENTROPY-REGULARIZED REINFORCE ALGORITHM

To train our policy network, we utilize a modified version of the REINFORCE algorithm proposed by Williams Williams [1992]. Our modifications include periodic updates of the policy network parameters, as opposed to updating them only at a fixed step limit $T$. This approach has been shown to improve the generalization of the policy network to larger values of $T$ during testing Zhang et al. [2024]. Additionally, to encourage exploration of the action space, we incorporate a regularization term $\mathcal{H}(\pi_\theta) = -\mathbb{E}_{a \sim \pi_\theta} \log(\pi_\theta(a))$ based on the entropy of the policy $\pi_\theta$ into the original objective of the REINFORCE algorithm. The complete learning procedure is outlined in Algorithm 1 in Appendix D.

## 5 EXPERIMENT

To comprehensively evaluate the performance of our TB-GAT, we conduct a series of experiments on both synthetic and publicly available datasets.

## 5.1 EXPERIMENTAL SETUP

The algorithm configurations can be found in Appendix E.

## 5.2 TESTING DATASETS AND BASELINES

**Datasets.** We evaluate the performance of our proposed method on two categories of datasets. The first category comprises synthetic datasets that we generated using the same method as the training dataset. The synthetic dataset includes five different sizes, namely $10 \times 10$, $15 \times 10$, $15 \times 15$, $20 \times 10$, and $20 \times 15$, each consisting of 100 instances. The second category includes seven widely used public benchmark datasets, i.e., Taillard Taillard [1993], ABZ Adams et al. [1988], FT Fisher [1963], LA Lawrence [1984], SWV Storer et al. [1992], ORB Applegate and Cook [1991], and YN Yamada and Nakano [1992]. These datasets contain instances with small and large scales, including those not seen during training, such as $100 \times 20$, which challenges the generalization ability of our model. It is worth noting that our model is trained with randomly generated synthetic datasets, whereas the seven open benchmark datasets are generated using distributions different from ours. Hence,

the results on these classic datasets can be considered the zero-shot generalization performance of our method. We test the model trained on the closest size for each problem size, e.g., the model trained on the size of $10 \times 10$ is used for testing on the problem of size $10 \times 10$ or others close to it.

**Baselines.** In order to demonstrate the superior performance of TBGAT, we conduct a comparative analysis against nine different baseline methods of various genres, including eight state-of-the-art neural approaches such as construction heuristics (L2D Zhang et al. [2020], RL-GNN Park et al. [2021b], ScheduleNet Park et al. [2021a], ACL Iklassov et al. [2022], JSSEnv Tassel et al. [2021], and DGERD Chen et al. [2022]), improvement heuristic (L2S Zhang et al. [2024]), and active search (EAS Kwon et al. [2021]). We also include an exact solver, CP-SAT Perron and Furnon, which has been shown to be robust and effective in solving JSSP Da Col and Teppan [2019] when given sufficient computational time (3600 seconds). For each problem size, we report the performance of our method in terms of the average relative gap to the best-known solutions, which are available online (for the seven classic benchmark datasets) or computed optimally with CP-SAT (for the synthetic evaluation dataset).[†] For the synthetic datasets, we compare our results against the optimal solution obtained using CP-SAT. The average relative gap is calculated by averaging the gap of each instance, which is defined as follows,

$$\sigma = (C_{max} - C_{max}^*)/C_{max}^* \times 100\%, \qquad (7)$$

where $C_{max}^*$ is the best-known solution (for the classic benchmark datasets) or the optimal solution (for the synthetic datasets).

## 5.3 PERFORMANCE ON PUBLIC BENCHMARKS

We present the results on public datasets. To present the evaluation results more clearly, we report the results for 500 improvement steps in Table 1 and the generalization results for different numbers of improvement steps in Table 2. In addition to the baselines mentioned earlier, we also include RL-GNN Park et al. [2021b] and ScheduleNet Park et al. [2021a] for comparison. The tables show that TB-GAT performs well when generalized to public benchmarks. Specifically, TBGAT achieves the best performance for all problem sizes and datasets, outperforming CP-SAT with a relative gap of 69.2% on Taillard $100 \times 20$ instances with a much shorter computational time of 6.7 minutes in Table 2, compared to 1 hour taken by CP-SAT. Moreover, TBGAT can find optimal solutions for several benchmark

---
[†]Please refer to http://optimizizer.com/TA.php and http://jobshop.jjvh.nl/.

datasets with different scales, such as $FT\ 6 \times 6$, $LA\ 15 \times 5$, $LA\ 20 \times 5$, and $LA\ 30 \times 10$, while L2S fails to do so. These results confirm that TBGAT achieves state-of-the-art results on the seven classic benchmarks and is relatively robust to different data distributions, as the instances in these datasets are generated using distributions substantially different from our training.

## 5.4 COMPARISON WITH OTHER SOTA BASELINES

Due to page limit, we leave the comparison against other SOTA baselines in Appendix G, including ACL Iklassov et al. [2022], JSSEnv Tassel et al. [2021], DGERD Chen et al. [2022]), and active search (EAS Kwon et al. [2021]).

## 5.5 ABLATION STUDY

We conducted an ablation study on the number of attention heads. We also empirically verify the linear computational complexity. Please refer to Appendix H for details.

## 6 CONCLUSION AND FUTURE WORK

We present a novel solution to the job shop scheduling problem (JSSP) using the topological-aware bidirectional graph attention neural network (TBGAT). Our method learns representations of disjunctive graphs by embedding them from both forward and backward views and utilizing topological sorts to enhance topological awareness. We also propose an efficient method to calculate the topological sorts for both views and integrate the TBGAT model into a local search framework for solving JSSP. Our experiments show that TBGAT outperforms a wide range of state-of-the-art neural baselines regarding solution quality and computational overhead. Additionally, we theoretically and empirically show that TBGAT possesses linear time complexity concerning the number of jobs and machines, which is essential for practical solvers.

## 7 ACKNOWLEDGEMENT

Wen Song is supported by the National Natural Science Foundation of China (Grant 62102228) and the Natural Science Foundation of Shandong Province (Grant ZR2021QF063). Zhiguang Cao is supported by the National Research Foundation, Singapore under its AI Singapore Programme (AISG Award No: AISG3-RP-2022-031), and the Singapore Ministry of Education (MOE) Academic Research Fund (AcRF) Tier 1 grant.

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

# Learning Topological Representations with Bidirectional Graph Attention Network for Solving Job Shop Scheduling Problem

**Cong Zhang**[1]   **Zhiguang Cao**[2]   **Yaoxin Wu**[3]   **Wen Song**[†,4]   **Jing Sun**[1]

[1]Nanyang Technological University, Singapore
[2]Singapore Management University, Singapore
[3]Department of Industrial Engineering & Innovation Sciences, Eindhoven University of Technology
[4]Institute of Marine Science and Technology, Shandong University, China
[1,4]{cong.zhang92@gmail.com, wensong@email.sdu.edu.cn}

## A   PROOF OF LEMMA 1

*Proof*. Since $O_{ji}$ is a prerequisite operation of $O_{mk}$, then there exists at least one path from $O_{ji}$ to $O_{mk}$ in the forward view of the disjunctive graph, i.e., $\mathcal{P}_{(O_{ji},O_{mk})} \neq \emptyset$, where $\mathcal{P}_{(O_{ji},O_{mk})}$ denotes the set of all paths from $O_{ji}$ to $O_{mk}$. Then, for any path $P(O_{ji},O_{mk}) \in \mathcal{P}_{(O_{ji},O_{mk})}$, by the transitivity of the topological sort, we can get that $\overrightarrow{\Phi}(O_{ji}) < \overrightarrow{\Phi}(O_{mk})$. Furthermore, because $\mathcal{P}_{(O_{ji},O_{mk})} \neq \emptyset$, we have $EST_{ji} < EST_{mk}$ due to the precedent constraints and the processing orders given by the disjunctive arcs, i.e., the operations at the head of the arcs always start earlier than that located at the tail of the arcs, which also transits from $O_{ji}$ to $O_{mk}$ by following the path $P(O_{ji},O_{mk})$. □

## B   PROOF OF COROLLARY 1

*Proof*. It is a similar procedure by following the proof of Lemma 1, but with each edge reversed.

## C   PROOF OF THEOREM 1

*Proof*. We show that $m_x^{L_x} = 1$ for all nodes $O_x \in \mathcal{O}$. First, it is obvious that $m_{\bar{x}}^{L_{\bar{x}}} = 1, \forall O_{\bar{x}} \in \bar{\mathcal{O}}$ since $m_{\bar{x}}^0 = 1$ and $\mathcal{N}_{\bar{x}} = \emptyset$. Second, one can prove that $m_{\tilde{x}}^{L_{\tilde{x}}} = 1$ by contradiction. Specifically, if $m_{\tilde{x}}^{L_{\tilde{x}}} \neq 1$, then there must exist a node $O_{\bar{x}'} \in \bar{\mathcal{O}}$ connecting to $O_{\tilde{x}}$ via a path, which has message $m_{\bar{x}'} \neq 0$, since $L_x$ is the maximum length. Hence, it contradicts with $m_{\bar{x}'}^0 = 1$. Next, if $L_x < L_z$ for any pair of nodes $O_x$ and $O_z$ connected by a path, it is clear that $\Phi(O_x) < \Phi(O_z)$ by the definition of the topological sort. □

---

[†]Corresponding author.

# D THE ENTROPY REGULARIZED REINFORCE ALGORITHM

---

**Algorithm 1** $n$-step REINFORCE

---

**Input**: Policy $\pi_\theta(a_t|s_t)$, step limit $T$, step size $n$, learning rate $\alpha$, training problem size $|\mathcal{J}| \times |\mathcal{M}|$, batch size $B$, total number of training instances $I$
**Output**: Trained policy $\pi_{\theta*}(a_t|s_t)$

1: **for** $i = 0$ to $i < I$ **do**
2:     Randomly generate $B$ instances of size $|\mathcal{J}| \times |\mathcal{M}|$, and compute their initial solutions $\{s_0^1, ..., s_0^B\}$ by using the dispatching rule FDD/MWKR
3:     Initialize a training data buffer $D^b$ with size 0 for each $s_0^b \in \{s_0^1, ..., s_0^B\}$;
4:     **for** $t = 0$ to $T$ **do**
5:         **for** $s_t^b \in \{s_t^1, ..., s_t^B\}$ **do**
6:             Compute a local move $a_t^b \sim \pi_\theta(a_t^b|s_t^b)$
7:             Update $s_t^b$ w.r.t $a_t^b$ and receive a reward $r(a_t^b, s_t^b)$
8:         **if** $t$ mod $n = 0$ **then**
9:             $loss_\theta^b = 0$
10:             **for** $j = n$ to $0$ **do**
11:                 $loss_\theta^b + = -(\log \pi_\theta(a_{t-j}^b|s_{t-j}^b) \cdot R_{t-j}^b + \mathcal{H}(\pi_\theta(a_{t-j}^b|s_{t-j}^b)))$, where $R_{t-j}^b$ is the return for step $t - j$ and $\mathcal{H}(\cdot)$ is the entropy;
12:             **end for**
13:             $\theta = \theta + \alpha \nabla_\theta \left( loss_\theta^b \right)$;
14:         **end if**
15:         **end for**
16:     **end for**
17:     $i = i + B$
18: **end for**
19: **return** $\pi_\theta(a_t|s_t)$

---

# E ALGORITHM CONFIGURATIONS

In our work, we generate random instances of JSSP as training data, across various sizes: small ($6 \times 6$, $10 \times 10$, $15 \times 10$), medium ($15 \times 15$, $20 \times 10$, $20 \times 15$), to large ($30 \times 10$, $30 \times 15$) scales, involving hundreds of operations. The operation processing times are assigned random values from 1 to 99, and the sequence of operations for each job is determined through random permutations.

The hyperparameters for our proposed TBGAT model are determined through empirical tuning on the small problem size of $10 \times 10$. For FEM and BEM, we utilize three layers with each layer consisting of four attention heads and a hidden dimension of 128 for both the input and hidden layers. The number of attention heads serves as the main parameter for our model. Therefore, we conduct an ablation study to investigate the correlation between the number of attention heads and performance (see the study in section H.1). Regarding the action selection network, $Net_A$ has $L_A = 4$ hidden layers, each with half of the dimensions of its parent layer. To ensure the stability of our training process, we normalize all raw features by dividing a large number. Specifically, we divide $\overrightarrow{\Phi}_{ji}$ and $\overleftarrow{\Phi}_{ji}$ by the largest sort $\overrightarrow{\Phi}^* = \max_{ji} \overrightarrow{\Phi}_{ji}$ and $\overleftarrow{\Phi}^* = \max_{ji} \overleftarrow{\Phi}_{ji}$, respectively. The processing time $p_{ji}$ is divided by 99, while $EST_{ji}$ and $LST_{ji}$ are both divided by 1000. Our model is trained for each problem size independently, with 128000 random instances uniformly distributed into 2000 batches of size 64, which are generated on the fly.

**Training phase configurations.** During training, we use $EC = 1e^{-5}$, $n = 10$, and $T = 500$ in our entropy-regularized REINFORCE algorithm with Adam optimizer and constant learning rate $lr = 1e^{-5}$. Throughout our experiments, actions are sampled from the policy. To ensure a fair comparison with L2S Zhang et al. [2024], the initial solutions are computed using the same priority dispatching rule $FDD/MWKR$ (minimum ratio of flow due date to most work remaining). Our TBGAT network is implemented in the Pytorch-Geometric (PyG)Fey and Lenssen [2019] framework. Other parameters follow the default settings in PyTorchPaszke et al. [2019]. We conduct all experiments on a workstation equipped with an

Table 3: Performance on synthetic datasets. "Gap": the average gap to the optimal solutions solved by CP-SAT. "Time": the average time for solving a single instance in seconds ("s") or minutes ("m"). For the neural methods, the best results are marked in bold, where the global best results are colored blue.

| Method | 10 × 10 Gap | 10 × 10 Time | 15 × 10 Gap | 15 × 10 Time | 15 × 15 Gap | 15 × 15 Time | 20 × 10 Gap | 20 × 10 Time | 20 × 15 Gap | 20 × 15 Time |
|---|---|---|---|---|---|---|---|---|---|---|
| CP-SAT Perron and Furnon | 0.0% | 0.3s | 0.0% | 8.2s | 0.0% | 2.6m | 0.0% | 3.9m | 0.0% | 45.7m |
| L2D Zhang et al. [2020] | 22.2% | 0.2 | 24.3% | 0.2s | 26.6% | 0.4s | 20.4% | 0.3s | 28.3% | 0.6s |
| L2S-500 Zhang et al. [2024] | 4.4% | 7.3s | 7.2% | 7.9s | 8.7% | 9.6s | 4.3% | 8.8s | 10.9% | 10.3s |
| TBGAT-500 | **2.7%** | 9.5s | **5.4%** | 11.2s | **6.7%** | 12.4s | **3.4%** | 12.8s | **9.3%** | 13.8s |
| L2S-1000 Zhang et al. [2024] | 3.5% | 15.0s | 5.7% | 16.0s | 7.9% | 18.7s | 3.4% | 17.3s | 9.6% | 21.3s |
| L2S-2000 Zhang et al. [2024] | 2.8% | 30.3s | 4.8% | 32.2s | 6.8% | 37.8s | 2.7% | 34.3s | 8.4% | 42.5s |
| L2S-5000 Zhang et al. [2024] | 2.2% | 75.2s | 4.0% | 80.9s | 5.9% | 91.9s | 2.3% | 85.1s | 7.2% | 1.6m |
| TBGAT-1000 | 2.1% | 18.4s | 4.3% | 22.0s | 5.6% | 24.2s | 2.4% | 24.9s | 7.7% | 26.8s |
| TBGAT-2000 | 1.7% | 35.9s | 3.7% | 44.4s | 4.8% | 47.5s | 2.0% | 49.1s | 6.5% | 52.8s |
| TBGAT-5000 | **1.4%** | 89.1s | **3.0%** | 1.8m | **4.1%** | 2.0m | **1.4%** | 2.0m | **5.6%** | 2.2m |

AMD Ryzen Threadripper 3960X 24-Core Processor and a single Nvidia RTX A6000 GPU.

**Evaluation phase configurations.** During testing, we employ the same hyperparameters as in the training phase and load the model with the optimal trained parameters. Furthermore, we evaluate the generalization performance of our model on larger improvement steps (up to 5000) since the effectiveness of the improvement heuristics heavily rely on sufficient search depth. Specifically, we train our model with $T = 500$ but evaluate its performance on 500, 1000, 2000, and 5000 improvement steps for each problem size, respectively.

# F  RESULTS ON SYNTHETIC DATASET

The experimental results on the synthetic datasets are summarized in Table 3, where the upper and lower decks display the results for basic testing (500 steps) and step-wise generalization (1000, 2000, and 5000 steps), respectively. From the results, we observe that L2S and TBGAT with 500 improvement steps achieve superior performance compared to L2D, primarily because improvement heuristics generally outperform construction ones in terms of solution quality. Furthermore, TBGAT outperforms L2S regarding solution quality, with consistently better results across all step horizons. In particular, TBGAT with 1000 steps achieves a smaller optimality gap than L2S with 2000 steps across all problem sizes. Moreover, TBGAT-5000 achieves the best overall results among all the compared neural methods. These findings suggest that TBGAT is more effective in learning representations for disjunctive graphs.

# G  COMPARE WITH MORE SOTA BASELINES

## G.1  COMPARISON WITH EAS

In this subsection, we present a comparison between TBGAT and EAS Kwon et al. [2021], a state-of-the-art active search method for solving JSSP. We evaluate the performance of both methods using instances of three different scales, ensuring a fair comparison with EAS. We also compare the three reported versions of EAS, namely EAS-Emb, EAS-Lay, and EAS-Tab. The results are presented in Table. 4. It is evident that TBGAT outperforms EAS with only 500 improvement steps, and TBGAT also has a significant advantage in computational time. This is because EAS is an active search method that requires additional time for fine-tuning on each problem instance, while TBGAT can quickly infer high-quality solutions once trained offline.

## G.2  COMPARISON WITH ACL

In this subsection, we compare TBGAT with ACL Iklassov et al. [2022], a curriculum learning method for learning priority dispatching rules that can generalize to different problem sizes, specifically on the Taillard dataset. Table. 5 presents the

Table 4: Performance compared with EAS. "Gap": the average gap to the optimal solutions solved by CP-SAT with 3600 seconds time limit.. "Time": the average time for solving a single instance in seconds ("s") or minutes ("m"). The best results are marked in bold, where the global best results are colored blue.

| Method | Synthetic Dataset | | | | | |
| --- | --- | --- | --- | --- | --- | --- |
| | $10 \times 10$ | | $15 \times 15$ | | $20 \times 15$ | |
| | Gap | Time | Gap | Time | Gap | Time |
| EAS-Emb Hottung et al. [2021] | 3.7% | 4.2m | 11.7% | 13.2m | 16.8% | 22.2m |
| EAS-Lay Hottung et al. [2021] | 6.5% | 4.2m | 13.8% | 15.0m | 17.6% | 27.6m |
| EAS-Tab Hottung et al. [2021] | 6.5% | 4.8m | 15.9% | 17.4m | 20.7% | 30.6m |
| TBGAT-500 | **2.7%** | 9.5s | **6.7%** | 12.4s | **9.3%** | 13.8s |
| TBGAT-1000 | 2.1% | 18.4s | 5.6% | 24.2s | 7.7% | 26.8s |
| TBGAT-2000 | 1.7% | 35.9s | 4.8% | 47.5s | 6.5% | 52.8s |
| TBGAT-5000 | **1.4%** | 81.9s | **4.1%** | 2.0m | **5.6%** | 2.2m |

Table 5: Performance compared with ACL. "Gap": the average gap to the best known solutions reported in the previous literature. The best results are marked in bold, where the global best results are colored blue.

| Method | Taillard | | | | | | | |
| --- | --- | --- | --- | --- | --- | --- | --- | --- |
| | $15 \times 15$ | $20 \times 15$ | $20 \times 20$ | $30 \times 15$ | $30 \times 20$ | $50 \times 15$ | $50 \times 20$ | $100 \times 20$ |
| | Gap | Gap | Gap | Gap | Gap | Gap | Gap | Gap |
| ACL Iklassov et al. [2022] | 15.0% | 17.7% | 17.4% | 20.4% | 21.9% | 15.7% | 16.0% | 9.6% |
| TBGAT-500 | **8.0%** | **9.9%** | **10.0%** | **13.3%** | **16.4%** | **9.6%** | **11.9%** | **6.4%** |
| TBGAT-1000 | 6.1% | 8.7% | 9.0% | 10.9% | 14.0% | 7.5% | 9.4% | 4.9% |
| TBGAT-2000 | 5.1% | 7.3% | 7.9% | 9.4% | 12.1% | 5.6% | 7.5% | 3.0% |
| TBGAT-5000 | **4.6%** | **6.3%** | **7.0%** | **6.8%** | **9.7%** | **2.6%** | **5.0%** | **1.2%** |

comparison results, which show that TBGAT-500 outperforms ACL by 36.4% on average optimality gap. The advantage of TBGAT-500 over ACL is further expanded when TBGAT-1000 and TBGAT-2000 are considered, demonstrating the effectiveness and robustness of TBGAT in solving JSSP instances of different scales.

## G.3 COMPARISON WITH JSSENV

Table 6: Detailed makespan values compared with the online DRL method JSSEnv on Taillard $30 \times 20$ dataset. Best results are marked as bold. "Time": the time for solving a single instance in seconds.

| Method | Taillard $30 \times 20$ Instances | | | | | | | | | | | | | | | | | | | | |
| --- | --- | --- | --- | --- | --- | --- | --- | --- | --- | --- | --- | --- | --- | --- | --- | --- | --- | --- | --- | --- | --- |
| | Tai-41 | | Tai-42 | | Tai-43 | | Tai-44 | | Tai-45 | | Tai-46 | | Tai-47 | | Tai-48 | | Tai-49 | | Tai-50 | | Gap |
| | Cmax | Time | Cmax | Time | Cmax | Time | Cmax | Time | Cmax | Time | Cmax | Time | Cmax | Time | Cmax | Time | Cmax | Time | Cmax | Time | % |
| JSSEnv Tassel et al. [2021] | **2208** | 600.0s | 2168 | 600.0s | 2086 | 600.0s | 2261 | 600.0s | 2227 | 600.0s | 2349 | 600.0s | 2101 | 600.0s | 2267 | 600.0s | 2154 | 600.0s | 2216 | 600.0s | 13.1 |
| TBGAT-5000 | 2255 | 173.7s | **2112** | 171.2s | **2059** | 170.9s | **2141** | 173.7s | **2173** | 172.7s | **2183** | 168.7s | **2051** | 166.1 | **2136** | 177.7s | **2130** | 176.4s | **2132** | 172.2s | 9.7 |

In this subsection, we further compare our method with JSSEnv Tassel et al. [2021], an online neural construction heuristic based on deep reinforcement learning only evaluated on Taillard $30 \times 20$ instances. As reported in its original paper, JSSEnv requires 600 seconds of solving time for each problem instance. In contrast, our method follows an offline training and online testing fashion, which saves significant time during evaluation. Regarding the performance in Table. 6, TBGAT almost outperforms JSSEnv on all instances except for 'Tai-41'. Importantly, JSSEnv learns to solve each instance online, which may be less efficient in generalizing to new unseen instances than our method, since TBGAT can be directly applied once trained offline.

## G.4 COMPARISON WITH DGERD

Table 7: Detailed makespan values compared with DGERD on Taillard instances. Best results are marked as bold.

| Method | Taillard Instances | | | | | | | | | | | | | | | |
| --- | --- | --- | --- | --- | --- | --- | --- | --- | --- | --- | --- | --- | --- | --- | --- | --- |
| | $15 \times 15$ | | $20 \times 15$ | | $20 \times 20$ | | $30 \times 15$ | | $30 \times 20$ | | $50 \times 15$ | | $50 \times 20$ | | $100 \times 20$ | |
| | Tai-01 | Tai-02 | Tai-11 | Tai-12 | Tai-21 | Tai-22 | Tai-31 | Tai-32 | Tai-41 | Tai-42 | Tai-51 | Tai-52 | Tai-61 | Tai-62 | Tai-71 | Tai-72 |
| DGERD Transformer Chen et al. [2022] | 1711.07 | 1639.30 | 1833.00 | 1765.07 | 2145.63 | 2015.89 | 2382.63 | 2458.52 | 2541.22 | 2762.26 | 3762.60 | 3511.20 | 3633.48 | 3712.30 | 6321.22 | 6232.22 |
| TBGAT-500 | **1333** | **1319** | **1532** | **1461** | **1805** | **1750** | **1995** | **2054** | **2384** | **2229** | **3139** | **3176** | **3190** | **3284** | **5885** | **5452** |
| TBGAT-1000 | 1333 | 1319 | 1532 | 1458 | 1805 | 1733 | 1949 | 2038 | 2315 | 2229 | 3052 | 3055 | 3094 | 3232 | 5814 | 5379 |
| TBGAT-2000 | 1333 | 1303 | 1532 | 1429 | 1805 | 1727 | 1903 | 2004 | 2281 | 2189 | 2969 | 2980 | 3042 | 3176 | 5748 | 5291 |
| TBGAT-5000 | **1314** | **1301** | **1476** | **1419** | **1776** | **1719** | **1842** | **1957** | **2255** | **2112** | **2869** | **2112** | **2978** | **3111** | **5631** | **5204** |

In this subsection, we continue to compare our method with DGERD Chen et al. [2022], another recent neural construction heuristic for solving JSSP. DGERD employs a GNN to learn the latent representations for constructing solutions from the partial solution represented with disjunctive graphs, which is similar to L2D. In the original experiment, they select several representative instances from the Taillard benchmarks for testing, where each instance is solved 50 times with DGERD, and the average makespan is reported. We evaluate the performance of our method and DGERD on the same instances, and the results are presented in Table 7. In general, our method with 500 improvement steps achieves a much smaller makespan than DGERD on each instance.

## H ABLATION STUDIES

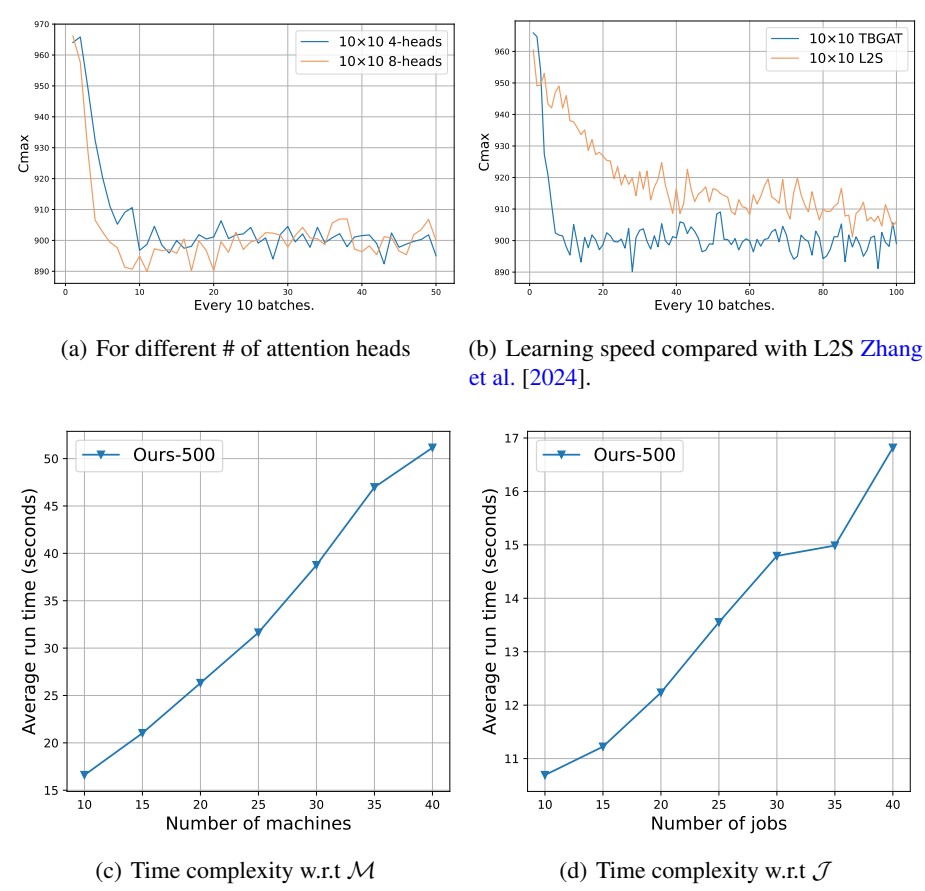

(a) For different # of attention heads

(b) Learning speed compared with L2S Zhang et al. [2024].

(c) Time complexity w.r.t $\mathcal{M}$

(d) Time complexity w.r.t $\mathcal{J}$

Figure 6: Ablation studies of the model architecture.

### H.1 THE NUMBER OF ATTENTION HEADS

The expressiveness of attention-based graph neural networks is known to depend on the number of attention heads. In general, each head learns problem representations independently, and emphasizes a different respective aspect of the problem that can compensate for each other, thus resulting in a more comprehensive understanding of the problem. Therefore, it is important to evaluate the impact of the number of heads in our TBGAT model. To this end, we conducted an ablation study, training TBGAT with four and eight attention heads on the $10 \times 10$ problem size, respectively. From the training curves in Fig. 6(a), we can observe that the TBGAT model with eight heads learns faster than the one with four heads, but the performance after convergence is similar. However, the eight-head TBGAT model appears to suffer from overfitting, as its performance starts to degrade after 100 batches of training.

### H.2 SAMPLING EFFICIENCY AGAINST L2S

To demonstrate the superiority of TBGAT over L2S Zhang et al. [2024] in terms of data efficiency during training, we present the learning curves of the two models in Fig. 6(b). As can be observed, TBGAT learns much faster than L2S, which can be attributed to its more appropriate graph embedding modules and more effective training algorithm (i.e., the entropy-regularized REINFORCE algorithm). This indicates that TBGAT is more sampling-efficient, i.e., requires fewer instances to reach a good performance level.

### H.3 VERIFICATION OF LINEAR COMPUTATIONAL COMPLEXITY

To verify Theorem 2, we examine the computational overhead of our method for solving instances of different scales. Fig. 6(c) shows the curves of computational time against the number of jobs $|\mathcal{J}|$ and machines $|\mathcal{M}|$, respectively. We can observe that our method exhibits roughly linear time complexity with respect to the number of jobs and machines. This finding supports Theorem 2, which states that the computational complexity of our method scales linearly with the problem size, making it practical for solving JSSP instances of large scales.