# OpenReview forum: "Learning Topological Representations with Bidirectional Graph Attention Network for Solving Job Shop Scheduling Problem"
_auai.org/UAI/2024/Conference — UAI 2024 poster_

### Official Review · Reviewer_L5Jx · 2024-03-19

**Q2-1 Originality-Novelty:** 3
**Q2-2 Correctness-Technical Quality:** 3
**Q2-5 Clarity Of Writing:** 3

**Q1 Summary And Contributions:**

This paper proposes a novel method for solving the Job Shop Scheduling Problem (JSSP), namely the Topology aware Bidirectional Graph Attention Network (TBGAT). The TBGAT model is the first bidirectional graph attention network designed specifically to address JSSP, capable of learning dependency relationships between tasks in both forward and backward dimensions. The experimental results indicate that TBGAT exhibits significant performance advantages in solving JSSP compared to existing SOTA methods.

**Q2-3 Extent To Which Claims Are Supported By Evidence:**

4: Excellent: all claims are supported by very convincing evidence (in the form of comprehensive experimental evaluation, rigorous mathematical proofs, detailed (pseudo-)code, precise references, well-motivated and realistic assumptions) and the authors deliver what they promise.

**Q2-4 Reproducibility:**

3: Good: key resources (e.g. proofs, code, data) are available and key details (e.g. proofs, experimental setup) are sufficiently well-described for competent researchers to confidently reproduce the main results.

**Q3 Main Strengths:**

1. The TBGAT model was proposed to learn the dependency relationships between tasks from both forward and backward perspectives, fully capturing the pre-constraints and resource allocation priorities of tasks in JSSP.
2. Introducing the concept of topological sorting adds rich structural information to the graph representation of JSSP, enabling the model to better understand the order and constraints between tasks.
3. Experiments conducted on multiple publicly available benchmark datasets have shown that TBGAT has achieved new optimal performance in multiple JSSP instances, surpassing several existing advanced methods.

**Q4 Main Weakness:**

The article mainly focuses on static JSSP, and for dynamic scheduling problems, TBGAT's response strategies are not discussed in detail, which is an important and common challenge in practical scheduling problems.

**Q5 Detailed Comments To The Authors:**

Although it has been demonstrated that TBGAT has linear computational complexity in theory, in practical applications, especially for large-scale problem instances, how to further optimize the computational efficiency of the algorithm, and reduce memory usage and computation time.

The authors should elaborate more clearly in the paper on the specific innovative points of TBGAT compared to existing methods, especially the unique role and contribution of bidirectional attention mechanism and topological sorting in the model. Further, it would be better to explain how these innovations overcome the limitations of previous methods and improve the performance and efficiency of JSSP solutions.

**Q9 Complying With Reviewing Instructions:**

Yes

---

> ### Author Rebuttal · Authors · 2024-04-06
>
> We thank the reviewer for posing constructive suggestions for future research and directions.
>
> **1. The article mainly focuses on static JSSP, and for dynamic scheduling problems, TBGAT's response strategies are not discussed in detail, which is an important and common challenge in practical scheduling problems.**
>
> **Responses**: We thank the reviewer for posing this interesting and challenging question. For the dynamic JSSP problem where dynamic events happen, such as the jobs arriving on the fly and the machines breaking down, solving the problem increases the complexity to another level, which requires the scheduling method to handle the dynamic events effectively. In industrial scenarios, dispatching rule-based methods are mostly used to address such dynamic scheduling problems [AR6]. However, the quality of the dispatching-rule-based methods is usually inferior to that of the improvement-based methods. Still, the latter is more suitable for static scheduling problems than the static JSSP tackled in this paper. Nonetheless, the proposed TBGAT remains effective for embedding the disjunctive graph. Therefore, we can integrate the TBGAT network with the dispatching strategy proposed in [AR7] to achieve dynamic scheduling.
>
> [AR6] Xiong, Hegen, et al. "A simulation-based study of dispatching rules in a dynamic job shop scheduling problem with batch release and extended technical precedence constraints." European Journal of Operational Research 257.1 (2017): 13-24.
>
> [AR7] Zhang, Cong, et al. "Learning to dispatch for job shop scheduling via deep reinforcement learning." Advances in neural information processing systems 33 (2020): 1621-1632.
>
> **2. How to further optimize the computational efficiency of the algorithm, and reduce memory usage and computation time.**
>
> **Responses**: This paper proposes an end-to-end learning-based improvement heuristic for solving JSSP. To make the entire algorithm efficient, we show that the proposed TBGAT has linear computational complexity regarding the number of jobs and machines (refer to Theorem 2 and Appendix H3 in the updated manuscript). To further improve the practical efficiency, we can leverage advanced parallel computing methods, e.g., [AR8]:
>
> [AR8] Der, Uwe, and Kathleen Steinhöfel. "A parallel implementation of a job shop scheduling heuristic." International Workshop on Applied Parallel Computing. Berlin, Heidelberg: Springer Berlin Heidelberg, 2000.
>
> We will investigate this direction for future research.
>
> **3. The authors should elaborate more clearly in the paper on the specific innovative points of TBGAT compared to existing methods, especially the unique role and contribution of bidirectional attention mechanism and topological sorting in the model. Further, it would be better to explain how these innovations overcome the limitations of previous methods and improve the performance and efficiency of JSSP solutions.**
>
> **Responses**: As delineated in the Introduction (3rd paragraph), the majority of extant learning-based methodologies for the Job Shop Scheduling Problem (JSSP), irrespective of whether they are constructive-based or improvement-based, utilize generic Graph Neural Network (GNN) models for graph embedding. The disjunctive graph representation of JSSP possesses a unique characteristic: the direction of the disjunctive arcs singularly determines a solution. In other words, the space of the directed disjunctive graph corresponds one-to-one with the feasible solution space. Therefore, a graph embedding network must encapsulate these topological structures inherent in the disjunctive graph, in conjunction with the scheduling features (such as processing time and commencement or completion time of the operations), to achieve optimal performance in solving JSSP problems. However, to the best of our knowledge, all existing GNN models overlook the topological sort features during the learning of graph embeddings. We introduce the TBGAT network to learn the topological features to bridge this gap. The experiment results show that the proposed network outperforms the existing SOTA leaning-based baselines by a large margin. In addition, we propose an MPTS method to enable efficient computation of topological sort features on the GPU, thereby significantly enhancing the training efficiency of our method.

---

### Official Review · Reviewer_tkEw · 2024-03-20

**Q2-1 Originality-Novelty:** 3
**Q2-2 Correctness-Technical Quality:** 3
**Q2-5 Clarity Of Writing:** 4

**Q1 Summary And Contributions:**

The paper proposes a novel GNN-based framework to solve the Job Shop Scheduling Problem (JSSP). The proposed framework better explores the inherent DAG structure in JSSP leveraging two GNNs with attention mechanisms, as well as an Action Selection Network (MLP) to determine schedule changes. The framework is evaluated in synthetic and benchmark datasets and compared to SOTA neural-based approaches, indicating its superiority.

**Q2-3 Extent To Which Claims Are Supported By Evidence:**

3: Good: the main claims are supported by convincing evidence (in the form of adequate experimental evaluation, proofs, (pseudo-)code, references, assumptions).

**Q2-4 Reproducibility:**

4: Excellent: key resources (e.g. proofs, code, data) are available and key details (e.g. proof sketches, experimental setup) are comprehensively described for competent researchers to confidently and easily reproduce the main results.

**Q3 Main Strengths:**

+ JSSP is a classic and difficult optimization problem that deserves to be addressed by novel neural-based methods.
+ Proposed approach incorporates the inherent structure of the JSSP (DAGs) into the framework.
+ Proposed framework is superior to prior and recent neural-based approaches to tackle JSSP.
+ Paer is well-written and clear to follow.

**Q4 Main Weakness:**

- Not clear how GNNs are trained. What is the loss function for the GNNs?
- Framework is trained for a given instance size (different instance sizes require different trained models).
- Results obtained by the framework are cannot surpass optimal results obtained via mathematical programming (CP-SAT).
- No discussion concerning the harder Flexible Job Shop Scheduling Problem (where jobs can run through a different set of machines, machines can be replicated, etc).

**Q5 Detailed Comments To The Authors:**

The proposed methodology is interesting because it explores more directly the inherent structure of DAGs in JSSP to build the framework, as opposed to recent neural-based approaches (as mentioned by the authors). The paper is also well-written and clear to understand. Some comments:
- How are the two GNNs trained? In particular, what is the loss function used by both FEM and BEM? This must be clearly described.
- Do you really need to train the model for different instance sizes?  Can the representation of the neighbors of a node (in the GNN) be averaged such that it does not grow with the size of instance?
- Claim about linear running time should be in the main paper and should be clearly stated that this is from empirical observations rather than theoretical analysis (running time complexity).
- It is a bit disappointing that the proposed framework cannot produce optimal results obtained by mathematical programming CP-SAT even on small instances when CP-SAT requires very little time. It would be interesting to understand why is this the case. Is there some other important inherent structure that CP-SAT is exploring that is not being captured by this neural-based framework?
- There is no discussion about the Flexible Job Shop Scheduling Problem (FJSSP) which is a more realistic problem formulation and also much harder to solve. It is possible that FJSSP is not as tough for the current framework as it is tough for mathematical programming approaches (specially for larger instances). But this can be left for future work!

**Q9 Complying With Reviewing Instructions:**

Yes

---

> ### Author Rebuttal · Authors · 2024-04-06
>
> We are grateful for the reviewer's recognition of our study.
>
> **1. How are the two GNNs trained? In particular, what is the loss function used by both FEM and BEM? This must be clearly described.**
>
> **Responses**: The detailed training procedure, including the input, output, and loss function, is clearly described in Algorithm 1 in Appendix D. Due to the page limit for the submission version, we have included Algorithm 1 as an appendix. As the final version of the paper will be two pages longer, we will move the training strategy to the main paper.
>
> We thank the reviewer for posing this suggestion, which will help us improve the quality of the manuscript.
>
> **2. Framework is trained for a given instance size (different instance sizes require different trained models).**
>
> **Responses**: We thank the reviewer for allowing us to answer this question. Even though we do not explicitly elaborate in the manuscript, our policy network can process a batch of disjunctive graphs with different problem sizes, i.e., the number of nodes, due to the GNN network that can naturally process a batch of graphs of various sizes, the proposed TBGAT network inherits such characteristics. The proposed MPTS operator also guarantees the calculation of topological features of graphs with different sizes within a batch, further facilitating batch processing. In summary, our proposed method is compatible with learning and testing on a batch of JSSP instances of diverse sizes.
>
> **3. It is a bit disappointing that the proposed framework cannot produce optimal results obtained by mathematical programming CP-SAT even on small instances when CP-SAT requires very little time. It would be interesting to understand why is this the case. Is there some other important inherent structure that CP-SAT is exploring that is not being captured by this neural-based framework?**
>
> **Responses**: Due to limited space, we kindly refer the reviewer to the response to reviewer **\#77C8** above for detailed explanations. The CP-SAT method is a constraint programming solver that uses SAT (satisfiability) methods for finding the optimal solutions, consuming a massive amount of time [AR4]. Our method is an improvement heuristic searching for high-quality solutions in a limited time. Therefore, CP-SAT is more suitable for instances that are not extremely large. In response to reviewer **\#77C8** above, we will show that our method outperforms CP-SAT by a large margin for extremely large instances (always witnessed in realistic manufacturing industries).
>
> [AR4] https://developers.google.com/optimization/cp
>
> **4. There is no discussion about the FlexibleJob Shop Scheduling Problem (FJSSP) which is a more realistic problem formulation and also much harder to solve, and this can be left for future work!**
>
> **Response**: We appreciate the opportunity to show that our method, with its disjunctive graph-based state representation and critical block-based neighbourhood structure, is adaptable to other scheduling problems like the flexible job-shop scheduling problem (FJSSP). The method requires slight adjustments for FJSSP, where the main complication is choosing among multiple machines for each operation. By tweaking our neighbourhood search heuristic to allow for the reassignment of operations to different eligible machines without forming cycles in the disjunctive graph, we can effectively address the FJSSP. Our approach involves selecting operations and their new positions based on defined neighbourhood structures, such as $Nopt_1$ and $Nopt_2$ in [AR5]. The modifications needed are mainly in the action set and state transition definitions of the Markov Decision Process (MDP) model. In contrast, the state, reward, and our proposed GNN model's effectiveness and computational efficiency remain the same due to the unchanged disjunctive graph representation (to see this, compare section 2 of the paper [AR5] with our disjunctive graph representation). We plan to explore extending our approach to FJSSP more thoroughly in future work.
>
> [AR5] Mastrolilli, M., & Gambardella, L. M. (2000). Effective neighborhood functions for the flexible job shop problem. Journal of scheduling, 3(1), 3-20.
>
> **5. Claim about linear running time should be clearly stated that this is from empirical observations.**
>
> **Responses**: We have added a Theorem in the updated version of the manuscript to theoretically prove that the proposed policy network (TBGAT network plus the action selection network) possesses linear computational complexity with respect to the number of jobs and machines respectively when computing the operation pair to be switched. In addition, since the proposed MPTS is based on the message-passing operator that can compute the topological features for all nodes in the disjunctive graph (Theorem 1) in parallel, the practical computational time for our method is linear to the problem sizes (Figure 6 of Appendix H.3).

---

### Official Review · Reviewer_77C8 · 2024-03-21

**Q2-1 Originality-Novelty:** 3
**Q2-2 Correctness-Technical Quality:** 3
**Q2-5 Clarity Of Writing:** 3

**Q1 Summary And Contributions:**

A new approach for applying machine-learning to solve the Job Shop scheduling problem (known to be NP-hard). The results show improvement.

**Q2-3 Extent To Which Claims Are Supported By Evidence:**

3: Good: the main claims are supported by convincing evidence (in the form of adequate experimental evaluation, proofs, (pseudo-)code, references, assumptions).

**Q2-4 Reproducibility:**

3: Good: key resources (e.g. proofs, code, data) are available and key details (e.g. proofs, experimental setup) are sufficiently well-described for competent researchers to confidently reproduce the main results.

**Q3 Main Strengths:**

A new successful approach for solving the Job Shop scheduling problem using machine-learning.

**Q4 Main Weakness:**

Not really a weakness of the work but the problem is that it still remains NP-hard. In some NP-hard problems the inputs appearing in practice practically always are solvable in polynomial time. But not here, I guess. The solution proposed has amazing speed but if I read the tables correctly, CP-SAT tends to produce better solutions.

**Q5 Detailed Comments To The Authors:**

In general, I find this piece of work good, though maybe not with a huge impact.

**Q9 Complying With Reviewing Instructions:**

Yes

---

> ### Author Rebuttal · Authors · 2024-04-06
>
> We appreciate the reviewer's acknowledgement of our research. We will address the main concern raised by the reviewer in the following.
>
> **1. The proposed solution has amazing speed, but if I read the tables correctly, I can see that CP-SAT tends to produce better solutions.**
>
> **Response**: The CP-SAT solver is known for its ability to find optimal solutions while sacrificing time efficiency (https://developers.google.com/optimization/cp). As a result, it is more suitable for smaller problem instances that do not require significant computational resources (like the instances for the public benchmarks employed in our paper, e.g., the maximal size is $100\times20$, which is far smaller than those in real manufacturing systems). On the other hand, our proposed heuristic improvement method is designed to quickly identify high-quality solutions within a realistic time frame, making it better suited for addressing complex industrial-scale problems.
>
> Specifically, we have demonstrated that our method surpasses CP-SAT by a significant margin when dealing with exceedingly large instances comprising at least 8000 operations. The results are showcased in the table below, where we imposed a one-hour time restriction on CP-SAT. The percentage indicates the difference in performance between our method and CP-SAT, with a negative percentage indicating our method's superiority. Our approach produces solutions of superior quality with much less time when faced with extremely large problem instances.
>
> |          |      200x40      |      500x60     |     1000x40     |
> |:--------:|:----------------:|:---------------:|:---------------:|
> |  CP-SAT  |    0.0% (1h)     |    0.0% (1h)    |    0.0% (1h)    |
> | TBGAT-500 | -25.11% (79.7s)  | -21.32% (3.2m)  | -15.99% (3.9m)  |

---

### Official Review · Reviewer_6hRG · 2024-03-21

**Q2-1 Originality-Novelty:** 2
**Q2-2 Correctness-Technical Quality:** 2
**Q2-5 Clarity Of Writing:** 3

**Q1 Summary And Contributions:**

The authors tackled the Job shop scheduling problem with a new graph neural network method. Their method uses the "topological-aware bidirectional graph attention neural network" TBGAT. The contribution is twofold: first, they propose to use two neural networks to take into account the forward and the backward view of the disjunctive graph ; second they define a new operator to propagate the message using the graph topology.
They performed experiments on both synthetic data and becnhmarks.

**Q2-3 Extent To Which Claims Are Supported By Evidence:**

3: Good: the main claims are supported by convincing evidence (in the form of adequate experimental evaluation, proofs, (pseudo-)code, references, assumptions).

**Q2-4 Reproducibility:**

3: Good: key resources (e.g. proofs, code, data) are available and key details (e.g. proofs, experimental setup) are sufficiently well-described for competent researchers to confidently reproduce the main results.

**Q3 Main Strengths:**

The main strength of this work is that the propose method performs well on the data shown by the authors.
The authors provide proofs and complementary results.

**Q4 Main Weakness:**

The main weakness of this paper is that there are some errors in the text, especially the one when they define the message-passing operator. Hopefully the proof (in appendix) is right and helped me understand.

**Q5 Detailed Comments To The Authors:**

If I am not mistaken, the Figure 1 has an error: the critical path is not a critical path since it's not the longest path from the source node to the sink node. There are two critical paths : OS - O31 - O21 - O22 - O11 - O12 - O13 - O23 - O33 - OT and OS - O31 - O32 - O22 - O11 - O12 - O13 - O23 - O33 - OT. The first path has one more critical block: O31 - O21, and the second path has a bigger critical block: O32 - O22 - O11.

Corollary 1 states that if Oji precedes Omk then LSTij > LSTmk. I am not sure about this, the latest starting time of Oij should be lower than the latest starting time of Omk if Oij should finish before Omk starts?

In the paragraph just below Corollary 1, the authors use "DRL" without defining it first. They do define every other acronym if the paper.

In the paragraph defining the message-passing operator, the authors states that m0 are initialized by 0 if the in-degree is 0 (which is, in the case of JSSP, only OS) and 1 otherwise. They state the contrary in the proof of Theorem 1. I agree with the proof and not with the text of the main paper.

In Figure 5, the exit of BEM is "Node Embedding from CAM" but CAM is not defined. Is it not "Node Embedding from BEM"?

In section 5.4 the authors refer to the appendix F instead of appendix G (which is currently not referred). They should introduce appendix F.

**Q9 Complying With Reviewing Instructions:**

Yes

---

> ### Author Rebuttal · Authors · 2024-04-06
>
> We really appreciate Reviewer \#6hRG's careful examination and detailed comments, which help us improve the manuscript's clarity. Here, we present our response to each comment.
>
> **1. Regarding the error in Figure 1.**
>
> **Response**: We thank the reviewer for pointing out this error. However, the definition here for 'longest' is not based on the number of nodes but on the total processing time of all operations along the path. Please refer to the quote ''The critical path (or paths) is the longest path (in time) from Start to Finish; it indicates the minimum time necessary to complete the entire project.'' in the following article for clarification:
>
> [AR3] https://hbr.org/1963/09/the-abcs-of-the-critical-path-method
>
> Therefore, according to the critical path definition, the path shown in Figure 1 (c) can indeed be a critical path, given that the total processing time of all the operations on the path is the largest among all paths from $O_S$ to $O_T$.
>
> We update the definition of the critical path to clarify any misleading information.
>
> Again, Thank you for the thorough review and finding of this subtle error.
>
> **2. Regarding the latest starting time $LST_{ij}$ and $LST_{mk}$ for operations $O_{ij}$ and $O_{mk}$ in Corollary 1.**
>
> **Responses**: Yes, you are correct. According to the definition of $LST$ in equation (2), it should be $LST_{ij} < LST_{mk}$. We will correct this mistake in the revised version of the paper.
>
>
> **3. In the paragraph below Corollary 1, the authors use "DRL" without defining it first.**
>
> **Responses**: We thank the reviewer for identifying this ill-defined abbreviation. We will add the meaning of the term 'DRL' in the revised manuscript.
>
> **4. Regarding the error in the paragraph defining the message-passing operator.**
>
> **Responses**: We thank the reviewer for identifying this mistake and will correct it in the main paper.
>
> **5. Regarding the naming error in Figure 5.**
>
> **Responses**: We thank the reviewer for identifying this mistake and will correct it in Figure 5.
>
> **6. Regarding the reference index error of Appendix F in Section 5.4.**
>
> **Responses**: We thank the reviewer for identifying this subtle indexing error. We have corrected the mistake accordingly.

---

### Official Review · Reviewer_EAgF · 2024-03-26

**Q2-1 Originality-Novelty:** 2
**Q2-2 Correctness-Technical Quality:** 3
**Q2-5 Clarity Of Writing:** 3

**Q1 Summary And Contributions:**

This work proposes the topology-aware bidirectional graph attention network (TBGAT), a GNN model based on the attention mechanism, to embed the DG for solving job shop schedule problem JSSP in a local search framework. Specifically, TBGAT embeds the DG from a forward and a backward view, respectively, where the messages are propagated by following the different topologies of the views and aggregated via graph attention. Message passing is then utilized to unify the topological features. Authors show empirical results of TBGAT against various neural methods.

**Q2-3 Extent To Which Claims Are Supported By Evidence:**

3: Good: the main claims are supported by convincing evidence (in the form of adequate experimental evaluation, proofs, (pseudo-)code, references, assumptions).

**Q2-4 Reproducibility:**

1: Poor: key details (e.g. proof sketches, experimental setup) are incomplete/unclear, or key resources (e.g. proofs, code, data) are unavailable.

**Q3 Main Strengths:**

Overall, the paper is well-written but because some important preliminaries and motivation are missing, it is difficult to understand the impact of the existing work.

The paper in general seems rigorous with corollaries and detailed proofs presented.

**Q4 Main Weakness:**

The last paragraph of the "Related Literature" is missing some recent works in the space of GNNs for graph attention, which the model uses for aggregating embeddings from topologies. GAT is a relatively old work and below are some more recent models that the authors should reference:

[WWW 2019] Heterogeneous Graph Attention Network. In The World Wide Web Conference (WWW '19). Association for Computing Machinery, New York, NY, USA, 2022–2032. https://doi.org/10.1145/3308558.3313562

[IEEE ICDM 2021] "Bi-Level Attention Graph Neural Networks," 2021 IEEE International Conference on Data Mining (ICDM), Auckland, New Zealand, 2021, pp. 1126-1131, doi: 10.1109/ICDM51629.2021.00133.

The authors need to provide more motivation as to why the JSSP problem is important to investigate, since the current writing does not reflect this.

**Q5 Detailed Comments To The Authors:**

Can the authors also more clearly indicate what the contributions (of novelty) for their work are?

**Q9 Complying With Reviewing Instructions:**

Yes

---

> ### Author Rebuttal · Authors · 2024-04-06
>
> We thank the reviewer for his or her time and effort in reviewing our paper. Here, we address each concern in order.
>
> **1. The reproducibility is poor.**
>
> **Response**: As mentioned in the 'Experiment Setup' section, Appendix E includes the detailed configurations, including hyperparameters for the proposed neural network, training, and testing configurations. We also guarantee that our code will be published once the paper is accepted.
>
> **2. The last paragraph of the "Related Literature" is missing some recent works in the space of GNNs for graph attention.**
>
> **Response**: We thank the reviewer for indicating the limitations of the literature review. We want to clarify that this paper's main contribution is to propose a novel end-to-end learning-based improvement heuristic method to solve the JSSP problem. The target is not to design a GNN structure that outperforms all recent GNN baselines. As an example in the following paper:
>
> [AR2] Ni, Fei, et al. "A multi-graph attributed reinforcement learning based optimization algorithm for large-scale hybrid flow shop scheduling problem." Proceedings of the 27th ACM SIGKDD Conference on Knowledge Discovery \& Data Mining. 2021.
>
> The literature review in [AR2] mainly focuses on approaches for scheduling problems rather than the GNN itself.
>
> Furthermore, the first paper referred by the reviewer:
>
> [WWW 2019] Heterogeneous Graph Attention Network. In The World Wide Web Conference (WWW '19). Association for Computing Machinery, New York, NY, USA, 2022–2032. https://doi.org/10.1145/3308558.3313562
>
> The proposed HAN network in this paper heavily relies on the random-walk-based sampling approaches for neighbourhood message aggregation, which is unsuitable for JSSP for the following reasons. For any disjunctive graph, each operation will have the predecessor and successor nodes modelling the precedent constraints and processing orders among machines. Sampling the neighbours via random walks will easily neglect any of the two aspects of information. Missing this critical information will lead to inferior performance.
>
> For the second literature pointed out by the reviewer:
>
> [IEEE ICDM 2021] "Bi-Level Attention Graph Neural Networks," 2021 IEEE International Conference on Data Mining (ICDM), Auckland, New Zealand, 2021, pp. 1126-1131, doi: 10.1109/ICDM51629.2021.00133.
>
> It is not guaranteed that the Bi-level attention GNN poses linear computational complexity as TBGAT proposed in our paper, which is an essential factor in evaluating a (neural-)solver for combinatorial optimization problems like JSSP.
>
> Nonetheless, we have added the two pieces of literature mentioned by the reviewer in the literature review.
>
> **3. The authors need to provide more motivation as to why the JSSP problem is important to investigate since the current writing does not reflect this.**
>
> **Response**: We appreciate the reviewer's perspective. The job shop scheduling problem (JSSP) is a widely recognized NP-hard combinatorial optimization dilemma with wide-ranging implications for contemporary manufacturing processes. Employing machine learning techniques, such as deep learning, to develop intelligent solutions for these intricate challenges holds great promise for propelling the smart manufacturing sector forward, garnering heightened interest. Although vehicle routing problems (VRP) in logistics have gained considerable traction, the job shop scheduling problem (JSSP), which is more intricate and imperative, tends to receive less attention. In this paper, we aim to lead the attention of the machine learning community to this challenging yet critical problem.
>
> The updated Introduction section in the revised manuscript will stress and state clearly the importance of investigating JSSP.
>
> **4. Can the authors also more clearly indicate the contributions (of novelty) of their work?**
>
> **Response**: We summarize the contributions as follows:
>
> 1. We propose a novel bidirectional graph attention network (TBGAT) tailored for disjunctive graphs, effectively capturing their unique topological features, which utilizes two independent graph attention modules to learn forward and backward views and incorporate forward and backward topological sorts.
>
> 2. We demonstrate that forward topological order in disjunctive graphs corresponds to global processing orderings and present an algorithm for efficient GPU computation.
>
> 3. We design an entropy-regularized REINFORCE algorithm to facilitate exploration during training effectively.
>
> 4. Theoretical analysis and experiments demonstrate that TBGAT’s linear computational complexity regarding the number of jobs and machines is a key attribute for practical JSSP solving.
>
> 5. According to comprehensive experimental results, we achieved new state-of-the-art (SOTA) performance across all datasets, significantly surpassing all neural baselines.

---

### Meta-Review · Area_Chair_Y7fg · 2024-04-17

For the pros, the paper is technically solid and has convincing experimental results on classic benchmarks (up to size 100x20).
For the cons, it compares an approximate method with an exact method (CP-SAT) which is able to prove optimality on small problems. The superiority of the approach compared to state-of-the-art, including CP-SAT or even better approaches such as CP Optimizer (see Teppan 2022 below), is not clearly established.

Industrial-size job shop scheduling with constraint programming
Giacomo Da Col, Erich C. Teppan, Operations Research Perspectives 2022.
https://doi.org/10.1016/j.orp.2022.100249

Minor remark: explain how you compute the relative gap of 69.2% (your absolute gaps in the Tables were already relative!)